# David and Goliath: Small One-step Model Beats Large Diffusion with Score Post-training

**Weijian Luo**[*][1]  **Colin Zhang**[1]  **Debing Zhang**[1]  **Zhengyang Geng**[2]

## Abstract

We propose *Diff-Instruct\*(DI\*)*, a data-efficient post-training approach for one-step text-to-image generative models to improve its human preferences without requiring image data. Our method frames alignment as online reinforcement learning from human feedback (RLHF), which optimizes the one-step model to maximize human reward functions while being regularized to be kept close to a reference diffusion process. Unlike traditional RLHF approaches, which rely on the Kullback-Leibler divergence as the regularization, we introduce a novel general score-based divergence regularization that substantially improves performance as well as post-training stability. Although the general score-based RLHF objective is intractable to optimize, we derive a strictly equivalent tractable loss function in theory that can efficiently compute its *gradient* for optimizations. We introduce *DI\*-SDXL-1step*, which is a 2.6B one-step text-to-image model at a resolution of $1024 \times 1024$, post-trained from DMD2 w.r.t SDXL. **Our 2.6B *DI\*-SDXL-1step* model outperforms the 50-step 12B FLUX-dev model** in ImageReward, PickScore, and CLIP score on the Parti prompts benchmark while using only 1.88% of the inference time. This result clearly shows that with proper post-training, the small one-step model is capable of beating huge multi-step diffusion models. Our model is open-sourced at this link: `https://github.com/pkulwj1994/diff_instruct_star`. We hope our findings can contribute to human-centric machine learning techniques.

[1]Hi Lab of Xiaohongshu Inc, Beijing, China. [2]School of Computer Science, Carnegie Mellon University, Pittsburgh, USA. Correspondence to: Weijian Luo <luoweijian@xiaohongshu.com>.

*Proceedings of the $42^{nd}$ International Conference on Machine Learning*, Vancouver, Canada. PMLR 267, 2025. Copyright 2025 by the author(s).

## 1. Introductions

Generative models have made substantial progress in recent years, largely improving the creative content creation across various domains (Karras et al., 2020; Nichol & Dhariwal, 2021; Poole et al., 2022; Kim et al., 2022; Tashiro et al., 2021; Meng et al., 2021; Couairon et al., 2022; Ramesh et al., 2022; Esser et al., 2024; Oord et al., 2016; Ho et al., 2022; Brooks et al., 2024; Zhang et al., 2023a; Xue et al., 2023; Luo & Zhang, 2024; Luo et al., 2023b; Pokle et al., 2022; Zhang et al., 2023b; Feng et al., 2023; Deng et al., 2024; Luo et al., 2024d; Geng et al., 2024; Wang et al., 2024).

Among these advancements, two types of generative models have gained significant attention, diffusion models (DMs) and one-step generators. Diffusion models (Sohl-Dickstein et al., 2015; Ho et al., 2020), or score-based generative models (Song et al., 2020), progressively corrupt data with diffusion processes and then train models to approximate the score functions of the noisy data distributions across varying noise levels. The learned score functions can generate high-quality samples by denoising the noisy samples through reversed stochastic differential equations. While DMs can produce good outputs, they often require a large number of model evaluations, which limits their efficiency in applications.

In contrast, one-step generators (Luo et al., 2024c;a; Zheng & Yang, 2024; Kang et al., 2023; Sauer et al., 2023a; Yin et al., 2024; Zhou et al., 2024a) have emerged as a highly efficient alternative to multi-step diffusion models. Unlike DMs, one-step generative models map latent noises directly to data samples in a single forward pass, making them highly efficient for real-time applications such as text-to-image and text-to-video generations on edge devices. Many existing works have demonstrated the leading performances of one-step text-to-image generators (Luo et al., 2024c; Zhou et al., 2024a; Yin et al., 2024) by employing diffusion distillation (Luo, 2023) as well as GAN techniques(Goodfellow et al., 2014; Brock et al., 2018; Karras et al., 2020; Sauer et al., 2023b).

- *Yet, these works only focus on matching the one-step model's distributions with pre-trained diffusion models or ground truth data, overlooking the critical*

*challenge of aligning one-step text-to-image models with human preferences, which is one of the central requirements of human-centric AI.*

To close this gap, we introduce *Diff-Instruct\** (**DI\***), a novel post-training approach that aligns one-step text-to-image generators with human preference while maintaining the ability to generate diverse and photo-realistic high-resolution images. We frame the human-preference alignment problem as reward maximization with score-based divergence constraints. This yields generated samples that not only have improved human reward but also adhere to user prompts. Unlike traditional RLHF methods (Christiano et al., 2017; Ouyang et al., 2022; Luo, 2024), which rely on Kullback-Leibler (KL) divergence for distribution regularization, we demonstrate that score-based regularization is important to preserve sample diversity and avoids reward hacking – the model generates weird, painting-like outputs with high rewards but low realism.

Our experiments demonstrate three key advantages: 1) *Efficiency*: The 2.6B *DI\*-SDXL-1step* model generates $1024 \times 1024$ resolution images in a single step, using only 1.88% of the inference time and 29.3% of the GPU memory of the 50-step 12B FLUX-dev model. 2) *Superior Performance*: On Parti prompts, *DI\*-SDXL-1step* achieves higher PickScore, ImageReward, and CLIPScore than FLUX-dev-50step. It also sets a new state-of-the-art HPSv2.1 score of 31.19 among open-source models. 3) *Fidelity*: The model maintains diversity and prompt adherence, matching FLUX-dev and SD3.5-large on COCO benchmarks while being orders of magnitude faster.

We summarize our contributions as follows:

- We introduce ***Diff-Instruct\**** for preference alignment of one-step text-to-image generative models with a strict theoretical guarantee based on a new score-based PPO paradigm;

- We introduce two novel approaches to incorporate classifier-free guidance into human preference alignment through explicit-implicit reward decoupling.

- With extensive evaluations, our best open-sourced 2.6B DI\*-SDXL-1step text-to-image model, a leading text-to-image model with a resolution of $1024 \times 1024$ that significantly outperforms the leading FLUX-dev-50step model with 1.88% inference time.

## 2. Preliminary

**Diffusion Models.** Here, we introduce preliminary knowledge and notations about diffusion models. Let $q_0(\boldsymbol{x}) = q_d(\boldsymbol{x})$ be the data distribution. The goal of generative modeling is to train models to generate new samples $\boldsymbol{x} \sim q_0(\boldsymbol{x})$. Under mild conditions, the forward diffusion

process of a diffusion model can transform initial distribution $q_0$ towards some simple noise distribution,

$$\mathrm{d}\boldsymbol{x}_t = \boldsymbol{F}(\boldsymbol{x}_t, t)\mathrm{d}t + G(t)\mathrm{d}\boldsymbol{w}_t, \qquad (2.1)$$

where $\boldsymbol{F}$ is a pre-defined vector-valued drift function, $G(t)$ is a pre-defined scalar-value diffusion coefficient, and $\boldsymbol{w}_t$ denotes an independent Wiener process. A continuous-indexed score network $\boldsymbol{s}_\varphi(\boldsymbol{x}, t)$ is employed to approximate marginal score functions of the forward diffusion process (2.1). The learning of score networks is achieved by minimizing a weighted denoising score matching objective (Vincent, 2011; Song et al., 2020),

$$\mathcal{L}_{DSM}(\varphi) = \int_{t=0}^{T} \lambda(t)\mathbb{E}_{\substack{\boldsymbol{x}_0 \sim q_0, \\ \boldsymbol{x}_t|\boldsymbol{x}_0 \sim p_t(\boldsymbol{x}_t|\boldsymbol{x}_0)}} \|\boldsymbol{s}_\varphi(\boldsymbol{x}_t, t)$$
$$- \nabla_{\boldsymbol{x}_t} \log p_t(\boldsymbol{x}_t|\boldsymbol{x}_0)\|_2^2 \mathrm{d}t.$$

The weighting function $\lambda(t)$ controls the importance of the learning at different time levels, and $p_t(\boldsymbol{x}_t|\boldsymbol{x}_0)$ denotes the conditional transition of the forward diffusion (2.1). After training, $\boldsymbol{s}_\varphi(\boldsymbol{x}_t, t) \approx \nabla_{\boldsymbol{x}_t} \log q_t(\boldsymbol{x}_t)$ is a good approximation of the marginal scores of the diffused data distribution.

## 3. Score-based Post-training

**Problem Setup.** Our basic setting is that we have a known human reward function $r(\boldsymbol{x}_0, \boldsymbol{c})$, which encodes the human preference for an image $\boldsymbol{x}_0$ and corresponding text description $\boldsymbol{c}$. Besides, we also have a pre-trained diffusion model which will later act as a reference distribution $p_{ref}(\boldsymbol{x}_0) = q_0(\boldsymbol{x}_0)$. The reference diffusion model is specified by the score function $\boldsymbol{s}_{q_t}(\boldsymbol{x}_t) := \nabla_{\boldsymbol{x}_t} \log q_t(\boldsymbol{x}_t)$, where $q_t(\boldsymbol{x}_t)$'s is the underlying distribution diffused at time $t$ according to (2.1). The pre-trained diffusion model represents the ground-truth data distribution that is used to prevent the one-step model from seeking high human reward but losing the ability to generate realistic images.

Our goal is to train a human-preferred one-step text-to-image model $g_\theta$ that generates images by directly mapping a random noise $\boldsymbol{z} \sim p_z$ to obtain $\boldsymbol{x}_0 = g_\theta(\boldsymbol{z}, \boldsymbol{c})$, conditioned on the input text $\boldsymbol{c} \sim \mathcal{C}$. We want the one-step model's output distribution, $p_\theta(\boldsymbol{x}_0|\boldsymbol{c})$, to maximize the expected human rewards while maintaining the ability to generate realistic images. We force $p_\theta(\boldsymbol{x}_0|\boldsymbol{c})$ to not diverge from $p_{ref}(\cdot)$ by using a divergence regularization term. Let $\boldsymbol{D}(\cdot, \cdot)$ be a distribution divergence. For any fixed prompt $\boldsymbol{c}$, the training objective is defined as:

$$\theta^* = \operatorname*{argmin}_\theta \mathbb{E}_{\boldsymbol{x}_0 \sim p_\theta(\boldsymbol{x}_0|\boldsymbol{c})} \Big\{ \big[ -\alpha r(\boldsymbol{x}_0, \boldsymbol{c}) \big] + \boldsymbol{D}(p_\theta, p_{ref}) \Big\} \quad (3.1)$$

Here $\alpha$ is a coefficient that balances reward influences and $\boldsymbol{D}(\cdot)$ acts as a regularization term. When using Kullback-Leibler divergence as the regularization term, the objective (3.1) turns to the online Proximal Policy Gradient (PPO) algorithm(Schulman et al., 2017), which has much success

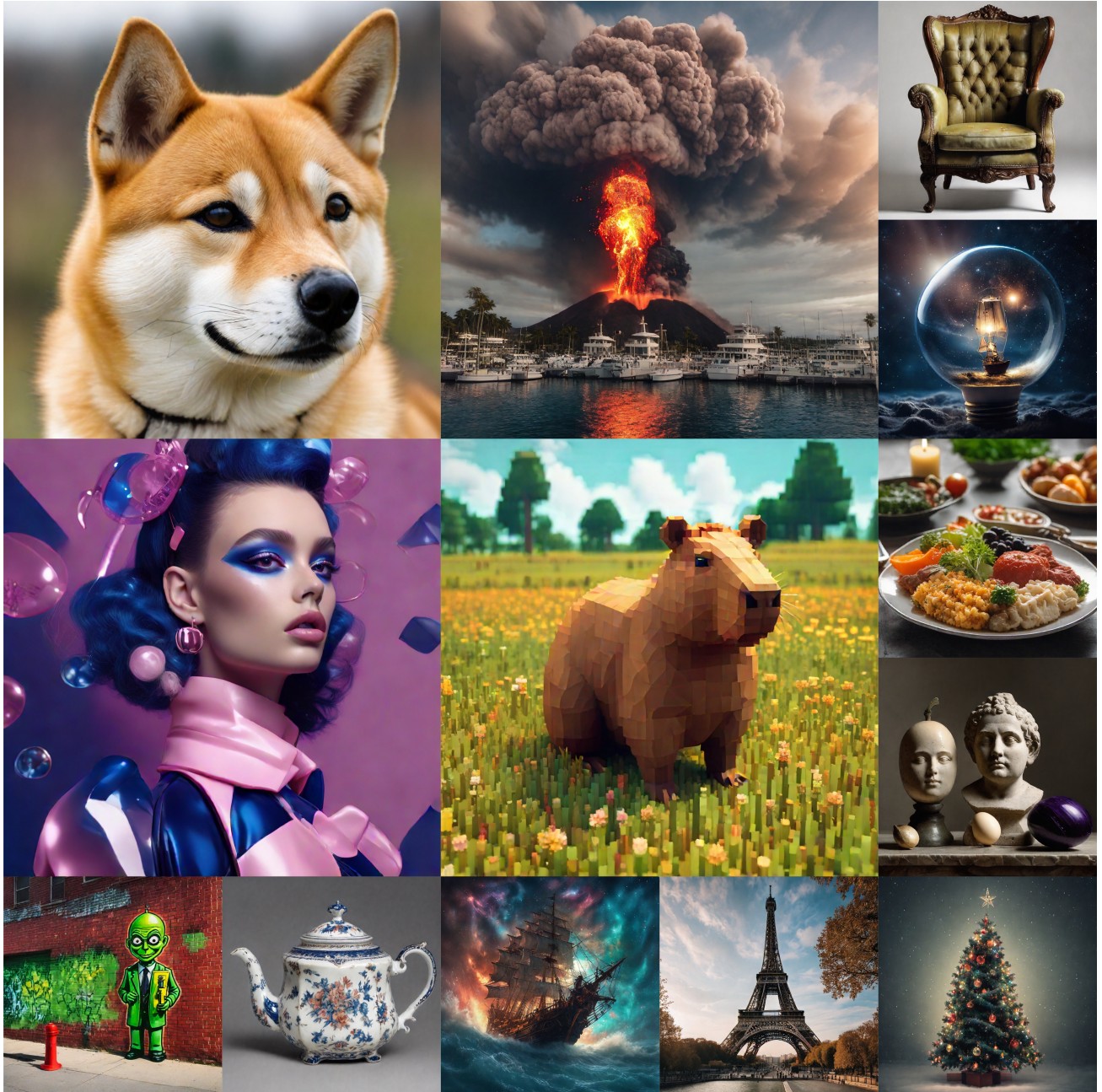

Figure 1: Generated $1024 \times 1024$ images from one-step 2.6B DI*-SDXL model. The DI*-SDXL-1step model outperforms FLUX-dev-50step(Lab, 2024) and SD3.5-large-28step(AI, 2024) in human preference metrics on the Parti prompt benchmark and HPSv2.1 scores. Please zoom in to check the details.

in reinforcement learning from human feedback (RLHF) for large language models (Ouyang et al., 2022) and one-step text-to-image models alignment (Luo, 2024). Particularly, recently (Luo, 2024) has shown that traditional KL-PPO can fairly train human-preferred one-step text-to-image models. However, KL divergence is notorious for mode-seeking behavior, which makes Luo (2024) very easy to collapse to some distribution modes – such as

painting-like images – that have high human reward but lack diversity. To address this issue, we find that general score-based divergences have an appealing diversity-preserving property that has been proved in one-step diffusion distillation (Luo et al., 2024c; Zhou et al., 2024b). Based on this intuition, we propose Diff-Instruct*, a novel post-training approach built upon a new Score-based online PPO paradigm that uses a general score-based divergence

to regularize the RLHF process.

## 3.1. Score-based Divergences

Inspired by Score-Implicit Matching (SIM (Luo et al., 2024c)) theory that revealed the advantages of general score-based divergences over the traditional KL divergence, we define the regularization term $\boldsymbol{D}(p_\theta, p_{ref})$ in (3.1) via the following general score-based divergence. Assume $\mathbf{d} : \mathbb{R}^d \to \mathbb{R}$ is a scalar-valued proper distance function (i.e., a non-negative function that satisfies $\forall \boldsymbol{x}, \mathbf{d}(\boldsymbol{x}) \geq 0$ and $\mathbf{d}(\boldsymbol{x}) = 0$ if and only if $\boldsymbol{x} = \boldsymbol{0}$). Given a parameter-independent sampling distribution $\pi_t$ that has large distribution support, we can formally define a time-integral score divergence as

$$\mathbf{D}^{[0,T]}(p_\theta, p_{ref}) := \int_0^T w(t) \mathbb{E}_{\pi_t} \mathbf{d}\big(\boldsymbol{s}_{p_{\theta,t}} - \boldsymbol{s}_{q_t}\big) \mathrm{d}t. \quad (3.2)$$

where $p_{\theta,t}$ and $q_t$ denote the marginal densities of the diffusion process (2.1) at time $t$ initialized with $p_{\theta,0} = p_\theta$ and $q_0 = p_{ref}$ respectively. $w(t)$ is an integral weighting function. Clearly, we have $\mathbf{D}^{[0,T]}(p_\theta, p_{ref}) = 0$ if and only if $p_\theta(\boldsymbol{x}_0) = p_{ref}(\boldsymbol{x}_0)$, $a.s.$ $\pi_0$.

## 3.2. Theories for Tractable Objectives

Recall that $g_\theta$ is a one-step model, therefore, samples from $p_\theta$ can be implemented through a direct and differentiable mapping $\boldsymbol{x}_0 = g_\theta(\boldsymbol{z}, \boldsymbol{c})$. With the score-based regularization term (3.2), for each given text prompt $\boldsymbol{c}$, we can formally write down our training objective to minimize as:

$$\mathcal{L}_{Orig}(\theta) = \mathbb{E}_{\substack{\boldsymbol{z} \sim p_z, \\ \boldsymbol{x}_0 = g_\theta(\boldsymbol{z}, \boldsymbol{c})}} \big[ -\alpha r(\boldsymbol{x}_0, \boldsymbol{c}) \big] + \mathbf{D}^{[0,T]}(p_\theta, p_{ref}) \quad (3.3)$$

Now we are ready to reveal the objective of Diff-Instruct* that we use to train human-preferred one-step models $g_\theta$. Notice that directly minimizing objective (3.3) is intractable because we do not know the relationship between $\theta$ and corresponding $p_{\theta,t}$. However, we show in Theorem 3.1 that an equivalent tractable loss (3.4) will have the same $\theta$ gradient as the intractable loss function (3.3):

$$\mathcal{L}_{DI*}(\theta) = \mathbb{E}_{\substack{\boldsymbol{z} \sim p_z, \\ \boldsymbol{x}_0 = g_\theta(\boldsymbol{z})}} \Big[ -\alpha r(\boldsymbol{x}_0, \boldsymbol{c}) \quad (3.4)$$
$$+ \int_{t=0}^T w(t) \mathbb{E}_{\substack{\boldsymbol{x}_t | \boldsymbol{x}_0 \\ \sim p_t(\boldsymbol{x}_t | \boldsymbol{x}_0)}} \Big\{ -\mathbf{d}'(\boldsymbol{y}_t) \Big\}^T$$
$$\Big\{ \boldsymbol{s}_{p_{\mathrm{sg}[\theta],t}}(\boldsymbol{x}_t) - \nabla_{\boldsymbol{x}_t} \log p_t(\boldsymbol{x}_t | \boldsymbol{x}_0) \Big\} \mathrm{d}t \Big].$$

with $\boldsymbol{y}_t := \boldsymbol{s}_{p_{\mathrm{sg}[\theta],t}}(\boldsymbol{x}_t) - \boldsymbol{s}_{q_t}(\boldsymbol{x}_t)$.

**Theorem 3.1.** Under mild assumptions, if we take the sampling distribution in (3.2) as $\pi_t = p_{\mathrm{sg}[\theta],t}$, then the gradient of (3.3) w.r.t $\theta$ is the same as (3.4):

$$\frac{\partial}{\partial \theta} \mathcal{L}_{Orig}(\theta) = \frac{\partial}{\partial \theta} \mathcal{L}_{DI*}(\theta).$$

In practice, we can use another assistant diffusion model $\boldsymbol{s}_\psi(\boldsymbol{x}_t, t)$ to approximate the one-step model's score func-

tion $\boldsymbol{s}_{p_{\mathrm{sg}[\theta],t}}(\boldsymbol{x}_t)$ pointwise, which was also done in the literature of diffusion distillations works such as Luo et al. (2024b;c); Zhou et al. (2024a;b); Yin et al. (2023; 2024). Therefore, we can alternate between 1) updating the assistant diffusion $\boldsymbol{s}_\psi(\boldsymbol{x}_t, t)$ using one-step model generated samples (which are efficient) and 2) updating the one-step model by minimizing the tractable objective (3.4). We name our training method that minimizes the objective $\mathcal{L}_{DI*}(\theta)$ in (3.4) the Diff-Instruct* because it is inspired by Diff-Instruct (Luo et al., 2024b) and Diff-Instruct++ (Luo, 2024) that involves an additional diffusion model and a reward model to train one-step text-to-image models.

## 3.3. Understanding Classifier-free Guidance

**Classifier-free Guidance Secretly Induced an Implicit Reward.** In previous sections, we have shown in theory that with explicitly available reward models, we can readily train the one-step models to align with human preference. In this section, we enhance the DI* by incorporating the classifier-free reward that is implied by the classifier-free guidance of diffusion models.

The classifier-free guidance (Ho & Salimans, 2022) (CFG) uses a modified score function of the form

$$\widetilde{\boldsymbol{s}}_{ref}(\boldsymbol{x}_t, t | \boldsymbol{c}) := \boldsymbol{s}_{ref}(\boldsymbol{x}_t, t | \varnothing) + \omega\{\boldsymbol{s}_{ref}(\boldsymbol{x}_t, t | \boldsymbol{c}) - \boldsymbol{s}_{ref}(\boldsymbol{x}_t, t | \varnothing)\},$$

to replace the original conditional score function $\boldsymbol{s}_{ref}(\boldsymbol{x}_t, t | \boldsymbol{c})$. Using CFG for diffusion models empirically leads to better sampling fidelity but with a cost of diversity.

As is first pointed out by Luo (2024), the classifier-free guidance is related to an implicit RLHF. In this part, we derive a tractable loss function that minimizes the so-called classifier-free reward, which we use together with the explicit reward $r(\cdot, \cdot)$ in DI*.

**Theorem 3.2.** Under mild conditions, if we set an implicit reward function as (3.6), the loss (3.5)

$$\mathcal{L}_{cfg}(\theta) = \int_0^T \mathbb{E}_{\substack{\boldsymbol{z} \sim p_z, \boldsymbol{x}_0 = g_\theta(\boldsymbol{z}, \boldsymbol{c}), \\ \boldsymbol{x}_t \sim p(\boldsymbol{x}_t | \boldsymbol{x}_0)}} w(t) \big[ \boldsymbol{s}_{ref}(\mathrm{sg}[\boldsymbol{x}_t] | t, \boldsymbol{c}) \quad (3.5)$$
$$- \boldsymbol{s}_{ref}(\mathrm{sg}[\boldsymbol{x}_t] | t, \varnothing) \big]^T \boldsymbol{x}_t \mathrm{d}t.$$

has the same gradient as the negative implicit reward function (3.6)

$$-r(\boldsymbol{x}_0, \boldsymbol{c}) = -\int_{t=0}^T \mathbb{E}_{\boldsymbol{x}_t \sim p_{\theta,t}} w(t) \log \frac{p_{ref}(\boldsymbol{x}_t | t, \boldsymbol{c})}{p_{ref}(\boldsymbol{x}_t | t)} \mathrm{d}t. \quad (3.6)$$

The notation $\mathrm{sg}[\boldsymbol{x}_t]$ means detaching the $\theta$ gradient on $\boldsymbol{x}_t$. Theorem 3.2 gives a tractable loss function (3.5) aiming to minimize the negative classifier-free reward function. Therefore, we can scale and add this loss $\mathcal{L}_{cfg}(\theta)$ (3.5) to the DI* loss (3.4) to balance the effects of explicit reward and implicit CFG reward.

**Choice 1: Incorporating CFG via Implicit CFG Reward.** Based on Theorem 3.2, we can incorporate

---

**Algorithm 1:** Diff-Instruct* Pseudo Code.

---

**Input:** prompt dataset $\mathcal{C}$, generator $g_\theta(\boldsymbol{x}_0|\boldsymbol{z}, \boldsymbol{c})$, prior distribution $p_z$, reward model $r(\boldsymbol{x}, \boldsymbol{c})$, reward model scale $\alpha_{rew}$, CFG reward scale $\alpha_{cfg}$, reference diffusion model $\boldsymbol{s}_{ref}(\boldsymbol{x}_t|c, \boldsymbol{c})$, assistant diffusion $\boldsymbol{s}_\psi(\boldsymbol{x}_t|t, \boldsymbol{c})$, forward diffusion $p(\boldsymbol{x}_t|\boldsymbol{x}_0)$ (2.1), assistant diffusion updates rounds $K_{TA}$, time distribution $\pi(t)$, diffusion model weighting $\lambda(t)$, generator IKL loss weighting $w(t)$.

**while** *not converge* **do**

  freeze $\theta$, update $\psi$ for $K_{TA}$ rounds by

  1. sample prompt $\boldsymbol{c} \sim \mathcal{C}$; sample time $t \sim \pi(t)$; sample $\boldsymbol{z} \sim p_z(\boldsymbol{z})$;

  2. generate fake data: $\boldsymbol{x}_0 = \mathrm{sg}[g_\theta(\boldsymbol{z}, \boldsymbol{c})]$; sample noisy data: $\boldsymbol{x}_t \sim p_t(\boldsymbol{x}_t|\boldsymbol{x}_0)$;

  3. update $\psi$ by minimizing loss: $\mathcal{L}(\psi) = \lambda(t)\|\boldsymbol{s}_\psi(\boldsymbol{x}_t|t, \boldsymbol{c}) - \nabla_{\boldsymbol{x}_t}\log p_t(\boldsymbol{x}_t|\boldsymbol{x}_0)\|_2^2$;

  freeze $\psi$, update $\theta$ using SGD:

  1. sample prompt $\boldsymbol{c} \sim \mathcal{C}$; sample time $t \sim \pi(t)$; sample $\boldsymbol{z} \sim p_z(\boldsymbol{z})$;

  2. generate fake data: $\boldsymbol{x}_0 = g_\theta(\boldsymbol{z}, \boldsymbol{c})$; sample noisy data: $\boldsymbol{x}_t \sim p_t(\boldsymbol{x}_t|\boldsymbol{x}_0)$;

  3. explicit reward: $\mathcal{L}_{rew}(\theta) = -\alpha_{rew}r(\boldsymbol{x}_0, \boldsymbol{c})$;

  4. CFG reward: $\mathcal{L}_{cfg}(\theta) = \alpha_{cfg} \cdot w(t)\{\boldsymbol{s}_{ref}(\mathrm{sg}[\boldsymbol{x}_t]|t, \boldsymbol{c}) - \boldsymbol{s}_{ref}(\mathrm{sg}[\boldsymbol{x}_t]|t, \varnothing)\}^T \boldsymbol{x}_t$;

  5. score-regularization: $\mathcal{L}_{reg}(\theta) = -w(t)\{\mathbf{d}'(\boldsymbol{s}_\psi(\boldsymbol{x}_t|t, \boldsymbol{c}) - \boldsymbol{s}_{ref}(\boldsymbol{x}_t|t, \boldsymbol{c}))\}^T \{\boldsymbol{s}_\psi(\boldsymbol{x}_t|t, \boldsymbol{c}) - \nabla_{\boldsymbol{x}_t}\log p_t(\boldsymbol{x}_t|\boldsymbol{x}_0)\}$;

  6. update $\theta$ by minimizing DI* loss: $\mathcal{L}_{DI*}(\theta) = \mathcal{L}_{rew}(\theta) + \mathcal{L}_{cfg}(\theta) + \mathcal{L}_{reg}(\theta)$;

**end**
**return** $\theta, \psi$.

---

classifier-free guidance by adding the pseudo reward (3.6) to explicit human reward to get a mixed reward. The guidance scale $\alpha_{cfg}$ and $\alpha_{rew}$ balances the strength of explicit human reward and implicit CFG reward.

**Choice 2: Incorporate CFG via CFG-enhanced Reference Diffusion.** Besides the implicit CFG reward, we can also inject the CFG mechanism into Diff-Instruct* by replacing the naive reference diffusion with a CFG-enhanced reference diffusion which writes $\widetilde{\boldsymbol{s}}_{ref}(\boldsymbol{x}_t|t, \boldsymbol{c}) := \boldsymbol{s}_{ref}(\boldsymbol{x}_t|t, \boldsymbol{c}) + \alpha_{cfg}(\boldsymbol{s}_{ref}(\boldsymbol{x}_t|t, \varnothing))$. This approach is straightforward. In practice, we find empirically that the **choice 2** is easier to tune and results in better performances. However, **choice 1** also leads to solid one-step models. Please check out ablation studies in Table 2 for details.

### 3.4. The Practical Algorithm

As Algorithm 1 shows, the DI* involves three models, with one one-step model $g_\theta$, one reference diffusion model $\boldsymbol{s}_{ref}$ and one assistant diffusion model $\boldsymbol{s}_\psi$. ***The reference diffusion is frozen, while the one-step model and the assistant diffusion are updated alternately.*** Two hyperparameters, the $\alpha_{rew}$ and $\alpha_{cfg}$, control the strength of the explicit reward and the implicit CFG reward during training. The explicit rewards can either be off-the-shelf reward models,

such as PickScore (Kirstain et al., 2023), CLIP score (Radford et al., 2021), or rewards trained in-house with internal feedback data.

**Pseudo-Huber Distance Functions.** Various choices of distance function $\mathbf{d}(.)$ result in different training algorithms. For instance, $\mathbf{d}(\boldsymbol{y}_t) = \|\boldsymbol{y}_t\|_2^2$ is a naive choice. This distance function has been studied in the pure diffusion distillation literature (Luo et al., 2024c; Zhou et al., 2024b;a). In this paper, we draw inspiration from (Luo et al., 2024c) and find that using the so-called pseudo-Huber distance leads to better performance. The distance writes $\mathbf{d}(\boldsymbol{y}) := \sqrt{\|\boldsymbol{y}_t\|_2^2 + c^2} - c$, and $\boldsymbol{y}_t := \boldsymbol{s}_{p_{\mathrm{sg}[\theta],t}}(\boldsymbol{x}_t) - \boldsymbol{s}_{q_t}(\boldsymbol{x}_t)$, then

$$\mathbf{D}^{[0,T]}(p_\theta, p_{ref}) \tag{3.7}$$
$$= -\left\{\frac{\boldsymbol{y}_t}{\sqrt{\|\boldsymbol{y}_t\|_2^2 + c^2}}\right\}^T \left\{\boldsymbol{s}_\psi(\boldsymbol{x}_t, t) - \nabla_{\boldsymbol{x}_t}\log p_t(\boldsymbol{x}_t|\boldsymbol{x}_0)\right\}.$$

## 4. Related Works

**Diffusion Distillation Through Divergence Minimization.** Diff-Instruct* is inspired by research on diffusion distillation (Luo, 2023), which aims to minimize distribution divergence to train one-step text-to-image models. Luo et al. (2024b) first studies the diffusion distillation by minimizing the Integral KL divergence. Yin et al. (2023) generalizes Diff-Instruct and adds a data regression loss for bet-

ter performance. Zhou et al. (2024b) study the distillation by minimizing the Fisher divergence. Luo et al. (2024c) study the distillation using the general score-based divergence. Many other works also introduced additional techniques and improved the performance (Geng et al., 2023; Kim et al., 2023; Song et al., 2023; Song & Dhariwal, 2024; Nguyen & Tran, 2023; Song et al., 2024; Yin et al., 2024; Zhou et al., 2024a; Heek et al., 2024; Xie et al., 2024; Salimans et al., 2024; Geng et al., 2024; Meng et al., 2022; Sauer et al., 2023b; Luo et al., 2023a; Liu et al., 2023; Gu et al., 2023; Xu et al., 2024; Ren et al., 2024; Lin et al., 2024; Zheng et al., 2024; Li et al., 2024; Berthelot et al., 2023; Zheng et al., 2022).

**Preference Alignment for Diffusion Models and One-step Models.** In recent years, many works have emerged trying to align diffusion models with human preferences. There are three main lines of alignment methods for diffusion models. 1) The first kind of method fine-tunes the diffusion model over a specifically curated image-prompt dataset (Dai et al., 2023; Podell et al., 2023). 2) The second line of methods tries to maximize some reward functions either through the multi-step diffusion generation output (Prabhudesai et al., 2023; Clark et al., 2023; Lee et al., 2023) or through policy gradient-based RL approaches (Fan et al., 2024; Black et al., 2023). For these methods, the backpropagation through the multi-step diffusion generation output is expensive and hard to scale. 3) The third line, such as Diffusion-DPO (Wallace et al., 2024), Diffusion-KTO (Yang et al., 2024), and MaPO (Hong et al., 2024) tries to directly improve the diffusion model's human preference property with raw collected data instead of reward functions. Besides the human preference alignment of diffusion models, Diff-Instruct++(Luo, 2024) recently arose as the first attempt to improve human preferences for one-step text-to-image models. Though we get inspiration from Diff-Instruct++, our Diff-Instruct* uses score-based divergences, which are technically different from the KL divergence used in DI++. Besides, as we show in Section 5.1, Diff-Instruct* outperforms Diff-Instruct++.

## 5. Experiments

**Experiment Settings for SDXL Experiment.** We use the open-sourced SDXL of $1024 \times 1024$ resolution as our reference diffusion in Algorithm 1. We construct the one-step model with the same architecture as the SDXL model and initialize the one-step model weights with the DMD2-SDXL-1step model (Yin et al., 2024), which is a good one-step text-to-image model distilled by using a combination of Diff-Instruct loss and GAN loss. We use the prompts of the LAION-AESTHETIC dataset with an aesthetic score larger than 6.25, resulting in a total of 2.2 million text prompts. We use the PickScore (Kirstain et al., 2023)(a

high-quality off-the-shelf reward model trained using human feedback data) as the explicit human reward. In Table 2, we do an ablation study that compares one-step DI* models with different guidance strategies and reward scales and finds that using inner guidance with a scale of 7.5 and an explicit human reward with a scale of 100 results in the best performance.

**Quantitative Evaluations Metrics.** We compare the DI*-SDXL-1step model with other leading open-sourced models of a resolution of $1024 \times 1024$ on the Parti prompt benchmark, COCO-2014-30K benchmark, and also Human preference scores. These models include diffusion-based (or flow-matching) models such as FLUX-dev (Lab, 2024), Stable-Diffusion-3.5-Large(AI, 2024), and SDXL (Podell et al., 2023) with its DPO variant (Wallace et al., 2024); 1024 resolution one-step models such as DMD2(Yin et al., 2024), Diff-Instruct(Luo et al., 2024b), Score Implicit Matching (SIM) (Luo et al., 2024c), and also Diff-Instruct++ (Luo, 2024). On Parti and COCO prompt datasets, we compute four standard scores in Table 1: the Image Reward (Xu et al., 2023), the Aesthetic Score (Schuhmann, 2022), the PickScore (Kirstain et al., 2023), and the CLIP score (Radford et al., 2021). We also evaluate the 30K-FID (Heusel et al., 2017) on the COCO validation dataset. On Human Preference Score (Wu et al., 2023) benchmarks, we calculate HPSv2.1 scores.

### 5.1. Performances and Findings

**2.6B DI*-SDXL-1step Model Shows Very Strong Performances.** The FLUX-dev-50step model (Lab, 2024) is recognized as a leading open-sourced diffusion model that is trained on millions of high-quality internal data samples. As shown in Table 1, our best 2.6B DI*-SDXL-1step model outperforms the 12B FLUX-dev-50step diffusion model and 8B SD3.5-large-28step using only 1.88% inference time and 29.3% GPU memory costs on the Parti prompt dataset which consists of challenging prompts with rich concepts. The 2.6B DI*-SDXL-1step is on par with FLUX and SD3.5 on COCO prompts in terms of preference scores and FIDs. As Table 3 shows, the DI*-SDXL-1step model has a record-breaking HPSv2.1 score of 31.19, outperforming the rest of the text-to-image models of $1024 \times 1024$ resolution. These empirical results confirm that proper post-training enables small one-step models to outperform huge multi-step models, analogous to the story of David and Goliath. See Figure 2 for visual comparisons with the 12B FLUX-Dev and the 8B Stable Diffusion 3.5-Large models.

**Ablation Study.** In Table 2, we conduct a comprehensive ablation study on the Parti prompt dataset to compare DI* (RLHF using score-based divergence) with different guidance strategies and reward scales (which recover previous distillation approaches such as Diff-Instruct(Luo et al., 2024b) and Score Implicit Matching(Luo et al., 2024c)),

Table 1: Quantitative comparisons of $1024 \times 1024$ resolution leading text-to-image models on **Parti(Yu et al., 2022)** Prompts (the upper part) and **MSCOCO-2014 validation** prompts (the under part). DI*: short for Diff-Instruct*. SIM: short for Score Implicit Matching. [†]: our implementation.

| MODEL | STEPS↓ | TYPE | PARAMS↓ | IMAGE↑ REWARD | AES↑ SCORE | PICK↑ SCORE | CLIP↑ SCORE | INFER TIME↓ PER 10 IMAGES |
|---|---|---|---|---|---|---|---|---|
| PARTI PROMPTS | | | | | | | | |
| SDXL-BASE(PODELL ET AL., 2023) | 50 | UNET | 2.6B | 0.887 | 5.72 | 0.2274 | 32.72 | 111 SEC |
| SDXL-DPO(WALLACE ET AL., 2024) | 50 | UNET | 2.6B | 1.102 | 5.77 | 0.2290 | **33.03** | 111 SEC |
| SD3.5-LARGE(AI, 2024) | 28 | DIT | 8B | **1.133** | 5.70 | 0.2306 | 32.70 | 66.23 SEC |
| FLUX-DEV(LAB, 2024) | 50 | DIT | 12B | 1.132 | **5.90** | **0.2317** | 31.70 | 118.64 SEC |
| DMD2-SDXL(YIN ET AL., 2024) | 1 | UNET | 2.6B | 0.930 | 5.51 | 0.2249 | 32.97 | 2.22 SEC |
| DIFF-INSTRUCT[†](LUO ET AL., 2024B) | 1 | UNET | 2.6B | 1.058 | 5.60 | 0.2253 | 33.02 | 2.22 SEC |
| SIM[†] (LUO ET AL., 2024C) | 1 | UNET | 2.6B | 1.049 | 5.66 | 0.2273 | 32.93 | 2.22 SEC |
| DIFF-INSTRUCT++-SDXL[†](LUO, 2024) | 1 | UNET | 2.6B | 1.061 | 5.58 | 0.2260 | 32.94 | 2.22 SEC |
| **DI*-SDXL (OURS)** | 1 | UNET | 2.6B | 1.067 | 5.74 | 0.2304 | 32.82 | 2.22 SEC |
| **DI*-SDXL (OURS, LONG TRAIN)** | 1 | UNET | 2.6B | 1.140 | 5.83 | 0.2331 | 32.75 | 2.22 SEC |
| **DI*-SDXL-DDPO (OURS, LONG TRAIN)** | 1 | UNET | 2.6B | **1.210** | **5.90** | **0.2342** | 32.91 | 2.22 SEC |
| COCO-2017-VAL 30K PROMPTS | | | | | | | | **COCO-FID↓** |
| SDXL-BASE(PODELL ET AL., 2023) | 50 | UNET | 2.6B | 0.825 | 5.55 | 0.2281 | 31.86 | **15.89** |
| SDXL-DPO(WALLACE ET AL., 2024) | 50 | UNET | 2.6B | 0.950 | **5.69** | 0.2295 | **32.17** | 21.48 |
| SD3.5-LARGE(AI, 2024) | 28 | DIT | 8B | 1.048 | 5.46 | 0.2305 | 31.85 | 17.69 |
| FLUX-DEV(LAB, 2024) | 50 | DIT | 12B | **1.065** | 5.64 | **0.2333** | 30.88 | 24.77 |
| DMD2-SDXL(YIN ET AL., 2024) | 1 | UNET | 2.6B | 0.822 | 5.42 | 0.2251 | 31.89 | **15.80** |
| DIFF-INSTRUCT-SDXL[†](LUO ET AL., 2024B) | 1 | UNET | 2.6B | 0.907 | 5.48 | 0.2264 | 31.85 | 21.62 |
| SIM-SDXL[†](LUO ET AL., 2024C) | 1 | UNET | 2.6B | 0.925 | 5.54 | 0.2277 | **31.90** | 18.84 |
| DIFF-INSTRUCT++-SDXL[†](LUO, 2024) | 1 | UNET | 2.6B | 0.908 | 5.53 | 0.2265 | 31.85 | 21.42 |
| **DI*-SDXL (OURS)** | 1 | UNET | 2.6B | 0.933 | 5.57 | 0.2305 | 31.75 | 19.09 |
| **DI*-SDXL (OURS, LONGER TRAINING)** | 1 | UNET | 2.6B | 0.983 | 5.65 | 0.2335 | 31.57 | 21.15 |
| **DI*-SDXL-DPO (OURS, LONG TRAIN)** | 1 | UNET | 2.6B | **1.034** | **5.65** | **0.2342** | 31.58 | 22.12 |

and DI++(Luo, 2024)(RLHF with KL divergence).

The results show that DI* using score-based divergence outperforms the DI++ model under the same configurations and reward scales. Notably, with proper reward scales, DI* models demonstrate superior human preference scores, indicating its simple yet effective contribution to aligning one-step text-to-image models with human feedback. Critically, DI++ with KL divergence tends to collapse to painting-like images with over-saturated color and lighting (Please refer to Figure 1 for visualizations), which lack diversity despite high rewards. Instead, score-based divergences empirically maintain diversity while steadily improving human preferences, suggesting that DI* can improve images' human preference scores regardless of their styles in paintings, photos, or animations.

In addition, we acknowledge the trade-off between preference scores and CLIP scores: increasing explicit human reward scales degrades CLIP scores. This phenomenon reveals a natural contradiction between human-preferred and objective data distributions. RLHF encourages the one-step model to generate human-preferred images, which inevitably shifts the underlying distribution from the ground-truth data distribution. This aligns with observations that FLUX, despite strong preference scores, underperforms in CLIPScore and FID compared to smaller models.

**DI* is compatible with Pre-aligned Diffusion Models** Beyond the ablation comparisons of different explicit reward strengths and CFG strengths, we also find that *DI* is compatible with pre-aligned reference diffusion models*. As Table 1 and Table 3 show, if we replace the reference diffusion model from SDXL to open-sourced SDXL-DDPO (Stable Diffusion XL with Diffusion Direct Preference Optimization (Wallace et al., 2024)), the preference alignment effects of the resulting one-step model are measured by PickScore, ImageReward, AestheticScore, and HPSv2.1, and were further improved. This demonstrates the compatibility and robustness of Diff-Instruct* with different reference diffusion models. In practice, we recommend that users use pre-aligned diffusion models that

**Qualitative Comparisons.** Figure 1 shows some $1024\times1024$ images generated by DI*-SDXL-1step models. Figure 2 shows a qualitative comparison of the DI*-SDXL-1step model against FLUX and SD3.5 diffusion models. The DI*-SDXL-1step model shows better aesthetic details, improved layouts, and reality lighting compared to FLUX and SD3.5, attributed to our score post-training. Figure 4 shows the visual comparison of DI* (with different ablative reward scales) against our initial model (the DMD2-1step model(Yin et al., 2024)), other one-step models, and SDXL with and without DPO. While SIM (Luo et al., 2024c), DI++(Luo, 2024), and Diff-Instruct(Luo et al., 2024b) can output high-quality

Table 2: Ablation study on **Parti(Yu et al., 2022)** Prompts of 1024 resolution DI*-SDXL-1step models with different reward scales. DI*: short for Diff-Instruct*. SIM: short for Score Implicit Matching. **DI*-Out**: incorporating CFG with implicit CFG reward; **DI*-In**: incorporating CFG with enhanced reference diffusion. [†]: our implementation.

| MODEL | STEPS | PARAMS | IMAGE↑ REWARD | AES↑ SCORE | PICK↑ SCORE | CLIP↑ SCORE | $(\alpha_{rev}, \alpha_{cfg})$ |
|---|---|---|---|---|---|---|---|
| DMD2-SDXL(INIT MODEL) | 1 | 2.6B | 0.938 | 5.51 | 0.2249 | 32.97 | − |
| DI++-SDXL[†] (ALIGNED USING KL) | 1 | 2.6B | 0.846 | 5.50 | 0.2243 | 32.66 | $(0, 0)$ |
| DI++-SDXL[†] (EQU TO DIFF-INSTRUCT) | 1 | 2.6B | 1.058 | 5.60 | 0.2253 | 33.02 | $(0, 7.5)$ |
| DI++-SDXL[†] (ALIGNED USING KL) | 1 | 2.6B | 1.061 | 5.58 | 0.2260 | 32.94 | $(100, 7.5)$ |
| **DI*-OUT-SDXL (OUT CFG)** | 1 | 2.6B | 1.082 | 5.63 | 0.2263 | **33.03** | $(100, 7.5)$ |
| **DI*-IN-SDXL (BASELINE, NO REWARD)** | 1 | 2.6B | 0.782 | 5.74 | 0.2256 | 32.16 | $(0, 0)$ |
| **DI*-IN-SDXL (EQU TO SIM, ONLY CFG)** | 1 | 2.6B | 1.049 | 5.66 | 0.2273 | 32.93 | $(0, 7.5)$ |
| **DI*-IN-SDXL (HUMAN REWARD + CFG))** | 1 | 2.6B | 1.031 | 5.69 | 0.2274 | 32.87 | $(1, 7.5)$ |
| **DI*-IN-SDXL** | 1 | 2.6B | 1.048 | 5.66 | 0.2278 | 32.91 | $(10, 7.5)$ |
| **DI*-IN-SDXL** | 1 | 2.6B | 1.020 | 5.68 | 0.2278 | 32.82 | $(100, 4.5)$ |
| **DI*-IN-SDXL** | 1 | 2.6B | 1.067 | 5.74 | 0.2304 | 32.82 | $(100, 7.5)$ |
| **DI*-IN-SDXL (BEST, LONGER TRAINING)** | 1 | 2.6B | **1.140** | **5.83** | **0.2331** | 32.75 | $(100, 7.5)$ |

Table 3: Quantitative evaluations of models on **HPSv2.1** scores. We compare open-sourced models regardless of their base model and architecture. † indicates our implementation.

| MODEL | ANIMATION↑ | CONCEPT-ART↑ | PAINTING↑ | PHOTO↑ | **AVERAGE↑** |
|---|---|---|---|---|---|
| 50STEP-SDXL-BASE(PODELL ET AL., 2023) | 30.85 | 29.30 | 28.98 | 27.05 | 29.05 |
| 50STEP-SDXL-DPO(WALLACE ET AL., 2024) | 32.01 | 30.75 | 30.70 | 28.24 | 30.42 |
| 28STEP-SD3.5-LARGE | 31.89 | 30.19 | 30.39 | 28.01 | 30.12 |
| 50STEP-FLUX-DEV | 32.09 | 30.44 | 31.17 | 29.09 | 30.70 |
| 1STEP-DMD2-SDXL(YIN ET AL., 2024) | 29.72 | 27.96 | 27.64 | 26.55 | 27.97 |
| 1STEP-DIFF-INSTRUCT-SDXL(LUO ET AL., 2024B) | 31.15 | 29.71 | 29.72 | 28.20 | 29.70 |
| 1STEP-SIM-SDXL(LUO ET AL., 2024C) | 31.97 | 30.46 | 30.13 | 28.08 | 30.16 |
| 1STEP-DI++-SDXL(LUO, 2024) | 31.19 | 29.88 | 29.61 | 28.21 | 29.72 |
| **1STEP-DI*-SDXL**(OURS) | 32.26 | 30.57 | 30.10 | 27.95 | 30.22 |
| **1STEP-DI*-SDXL**(OURS, LONGER TRAINING) | 33.22 | 31.67 | 31.25 | 28.62 | 31.19 |
| **1STEP-DI*-SDXL-DDPO**(OURS, LONGER TRAINING) | **33.92** | **32.80** | **32.71** | **29.62** | **32.26** |

samples, they suffer from oversaturation issues, meaning that the images have over-saturated colors and lighting that might make users feel uncomfortable. On the contrary, the DI*-1step model shows very gentle lighting and aesthetic colors, which align better with human preferences. We empirically conclude that DI* prevents the model from collapsing into high-reward painting-like images, in which DI++ collapsed. Besides, the SDXL and SDXL-DPO are easy to generate painting-like images instead of photo-realistic images.

## 6. Conclusion and Limitations

In this paper, we present Diff-Instruct*, a novel approach for preference alignment of one-step text-to-image generative models. We introduce two novel approaches to incorporate traditional classifier-free guidance into alignment through the lens of explicit-implicit reward decoupling. Using DI*, we introduce the 2.6B DI*-SDXL-1step model that outperforms current leading models such as the 12B FLUX-dev-50step model with 1.88% inference cost.

Nonetheless, Diff-Instruct* has its limitations. We empir-

ically found that flaws commonly observed in other one-step models also appeared in Diff-Instruct*. For example, the one-step model sometimes generates inaccurate human faces and hands. We believe that these bad cases are caused by the limitations of the one-step generation approach, which poses challenges for the model in generating correct content in a single pass. We believe that generalizing Diff-Instruct* to multi-step (few-step) models could help combine the advantages of both efficiency and generation correctness. Our results suggest that consistently improving the one-step model architecture, scaling its parameters, improving reference diffusions, and enhancing the reward models can lead to better models.

Finally, we identify directions for further research for score post-training. For instance, while DI* needs a pre-trained reward model, methods like DPO (Rafailov et al., 2024; Wallace et al., 2024) have introduced aligning generative models directly using human feedback data. It would be valuable to explore efficient score post-training for one-step generative models by directly incorporating feedback data and eliminating the pre-trained reward models.

DI*-SDXL-1step (2.6B)  FLUX-Dev-50step (12B)  SD3.5-large-28step (8B)

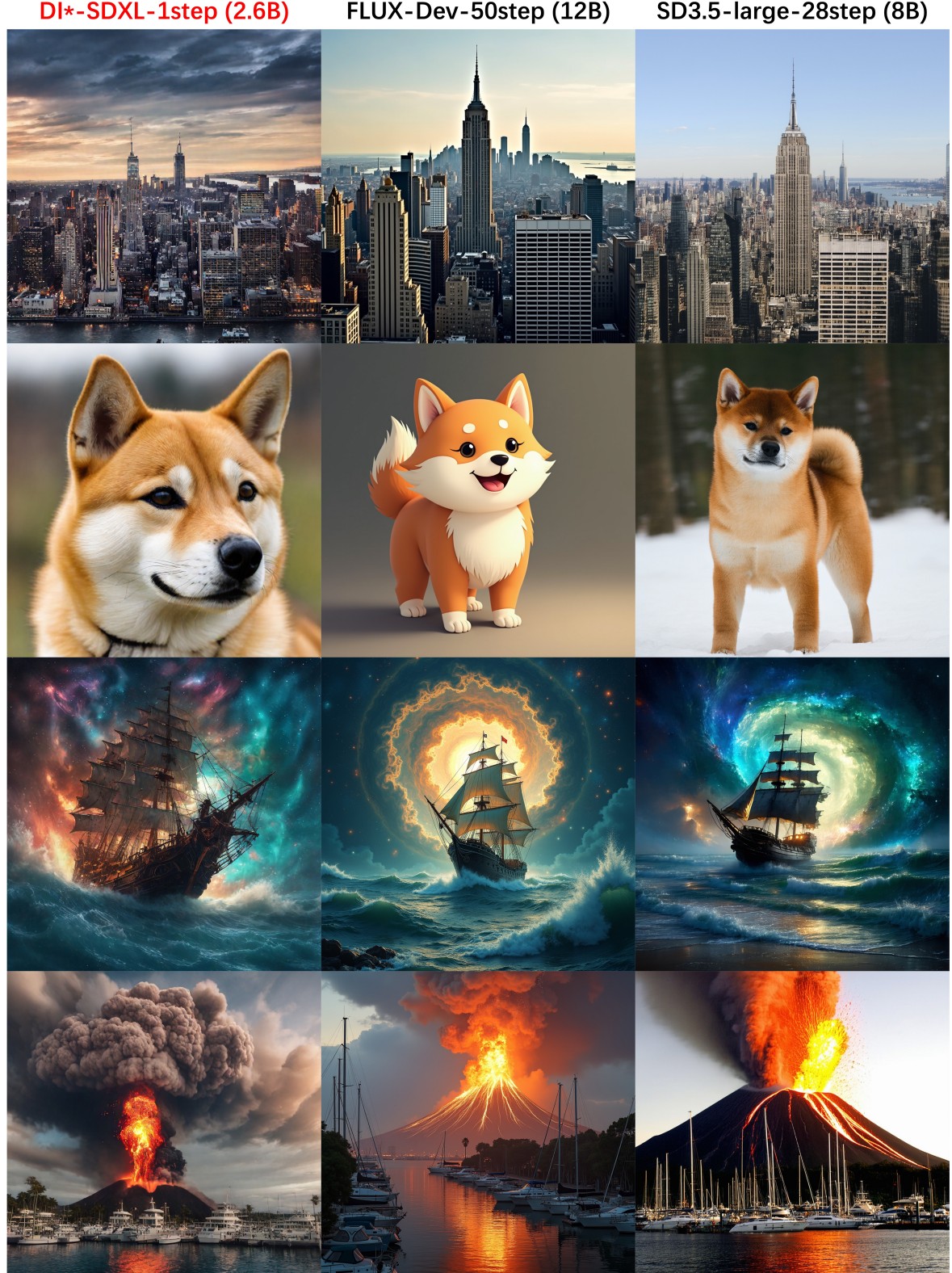

Figure 2: Comparison on Parti prompts of our **2.6B DI*-SDXL-1step** model against 12B FLUX-dev-50step and 8B SD3.5-large-28step diffusion models. Prompts used are listed in the appendix.

## Acknowledgement

We acknowledge constructive suggestions and feedback from reviewers and ACs/SACs/PCs of NeurIPS 2024. We thank the authors of DMD2 for their open-weight DMD2 models, which we use as the base model before post-training.

## Impact Statement

We propose Diff-Instruct* (DI*) that can effectively post-train one-step text-to-image generative models to be aligned with human preferences. The method and corresponding aligned models lie in the domain of Reinforcement Learning using Human Feedback, which involves the participation of humans for training reward models. Though the Diff-Instruct* algorithm itself does not generate harms to society, any misuse of the algorithm could potentially lead to misleading, fake, or biased generated content. We conduct experiments on academic benchmarks, whose resulting models are less likely to be misused. Further experiments are needed to better understand these consistency model limitations and propose solutions to address them

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

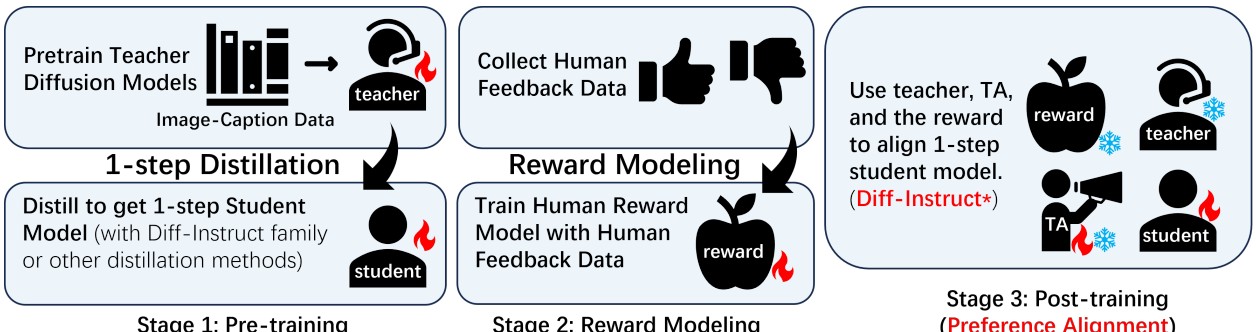

Figure 3: Workflow of developing one-step text-to-image generative models using Diff-Instruct*. The workflow includes stages of pre-training, reward modeling, and preference alignment.

## A. Theoretical Results

### A.1. Proof of Theorem 3.1

*Proof.* Recall that $p_\theta(\cdot)$ is induced by the one-step model $g_\theta(\cdot)$, therefore the sample is obtained by $\boldsymbol{x}_0 = g_\theta(\boldsymbol{z}|\boldsymbol{c}), \boldsymbol{z} \sim p_z$. The term $\boldsymbol{x}$ contains parameter through $\boldsymbol{x}_0 = g_\theta(\boldsymbol{z}|\boldsymbol{c}), \boldsymbol{z} \sim p_z$. To demonstrate the parameter dependence, we use the notation $p_\theta(\cdot)$. Note that $p_{ref}(\cdot)$ is the reference distribution. The alignment objective writes

$$\mathcal{L}_{Orig}(\theta) = \mathbb{E}_{\substack{\boldsymbol{z} \sim p_z, \\ \boldsymbol{x}_0 = g_\theta(\boldsymbol{z}|\boldsymbol{c})}} \big[ -\alpha r(\boldsymbol{x}_0, \boldsymbol{c}) \big] + \mathbf{D}^{[0,T]}(p_\theta, p_{ref}) \tag{A.1}$$

The first loss term of (A.1) $\alpha r(\boldsymbol{x}_0, \boldsymbol{c})$ is easy to compute by directly pushing the generated sample $\boldsymbol{x}_0$ and the text prompt $\boldsymbol{c}$ into the reward model $r(\cdot, \cdot)$. However, the second loss term (A.2) is intractable because we do not explicitly know the relation between $\theta$ and $p_{\theta,t}(\cdot)$.

$$\mathbf{D}^{[0,T]}(p_\theta, p_{ref}) \coloneqq \int_{t=0}^{T} w(t) \mathbb{E}_{\boldsymbol{x}_t \sim \pi_t} \bigg\{ \mathbf{d}(\boldsymbol{s}_{p_{\theta,t}}(\boldsymbol{x}_t) - \boldsymbol{s}_{q_t}(\boldsymbol{x}_t)) \bigg\} \mathrm{d}t, \tag{A.2}$$

We turn to derive the equivalent loss for $\mathbf{D}^{[0,T]}(p_\theta, p_{ref})$. First we take the $\theta$ gradient of (A.2), show

$$\frac{\partial}{\partial \theta} \mathbf{D}^{[0,T]}(p_\theta, p_{ref}) = \frac{\partial}{\partial \theta} \int_{t=0}^{T} w(t) \mathbb{E}_{\boldsymbol{x}_t \sim \pi_t} \bigg\{ \mathbf{d}(\boldsymbol{s}_{p_{\theta,t}}(\boldsymbol{x}_t) - \boldsymbol{s}_{q_t}(\boldsymbol{x}_t)) \bigg\} \mathrm{d}t \tag{A.3}$$

$$= \mathbb{E}_{t, \boldsymbol{x}_t \sim \pi_t} w(t) \bigg\{ \mathbf{d}'(\boldsymbol{y}_t) \bigg\}^T \frac{\partial}{\partial \theta} \boldsymbol{s}_{p_{\theta,t}}(\boldsymbol{x}_t) \tag{A.4}$$

Notice that $p_{\theta,t}(\cdot)$ is induced by first generating samples with one-step one-step model then adding noise with diffusion process (2.1), we do not know the term $\frac{\partial}{\partial \theta} \boldsymbol{s}_{p_{\theta,t}}(\boldsymbol{x}_t)$. Therefore the gradient formula (A.4) is intractable. However, we will show that a tractable loss function can recover the intractable gradient (A.4), and therefore can be used for minimizing (A.2). Our proof is inspired by the theory from Vincent (2011), Zhou et al. (2024b) and Luo et al. (2024c).

We first present a so-called Score-projection identity (Theorem A.1), which has been studied in Zhou et al. (2024b) and Vincent (2011):

**Theorem A.1.** Let $\boldsymbol{u}(\cdot)$ be a $\theta$-free vector-valued function under mild conditions, the identity holds:

$$\mathbb{E}_{\substack{\boldsymbol{x}_0 \sim p_{\theta,0}, \\ \boldsymbol{x}_t|\boldsymbol{x}_0 \sim q_t(\boldsymbol{x}_t|\boldsymbol{x}_0)}} \boldsymbol{u}(\boldsymbol{x}_t)^T \bigg\{ \boldsymbol{s}_{p_{\theta,t}}(\boldsymbol{x}_t) - \nabla_{\boldsymbol{x}_t} \log q_t(\boldsymbol{x}_t|\boldsymbol{x}_0) \bigg\} = 0, \quad \forall \theta. \tag{A.5}$$

We give a short proof of Theorem A.1 as a clarification. Readers can also refer to Vincent (2011) or Zhou et al. (2024b) as a reference.

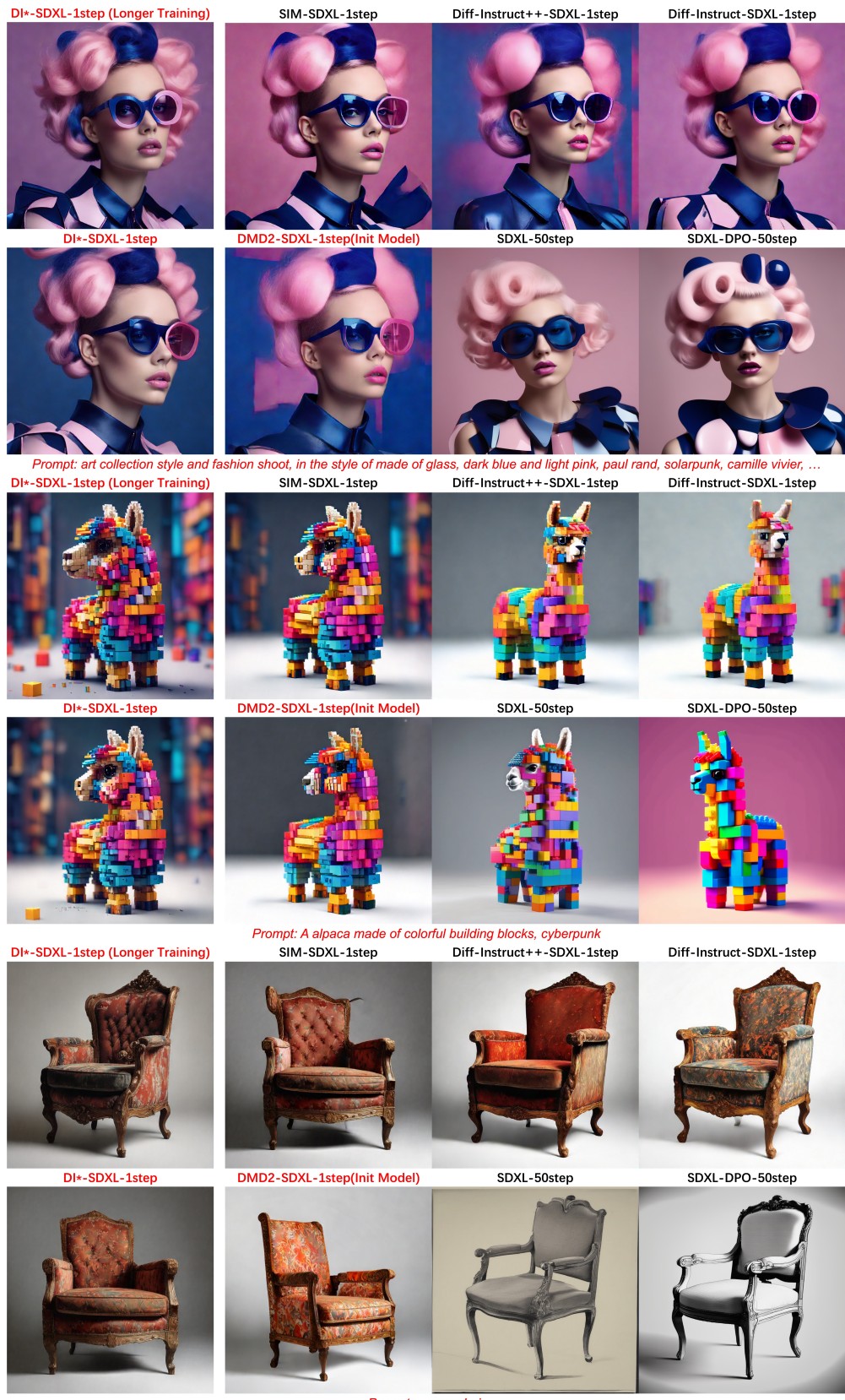

Figure 4: Ablation comparisons of DI*-1step models and other SDXL-based models in Table 2.

Recall the relation between $\boldsymbol{s}_{p_{\theta,t}}(\boldsymbol{x}_t)$ and $\nabla_{\boldsymbol{x}_t} \log q_t(\boldsymbol{x}_t|\boldsymbol{x}_0)$, we know

$$\boldsymbol{s}_{p_{\theta,t}}(\boldsymbol{x}_t) = \nabla_{\boldsymbol{x}_t} \log \int p_{\theta,0}(\boldsymbol{x}_0) q_t(\boldsymbol{x}_t|\boldsymbol{x}_0) \mathrm{d}\boldsymbol{x}_0$$

$$= \frac{\int p_{\theta,t}(\boldsymbol{x}_0) \nabla_{\boldsymbol{x}_t} q_t(\boldsymbol{x}_t|\boldsymbol{x}_0) \mathrm{d}\boldsymbol{x}_0}{p_{\theta,t}(\boldsymbol{x}_t)}$$

$$= \int \frac{\nabla_{\boldsymbol{x}_t} \log q_t(\boldsymbol{x}_t|\boldsymbol{x}_0) p_{\theta,t}(\boldsymbol{x}_0) q_t(\boldsymbol{x}_t|\boldsymbol{x}_0)}{p_{\theta,t}(\boldsymbol{x}_t)} \mathrm{d}\boldsymbol{x}_0$$

We have

$$\mathbb{E}_{\substack{\boldsymbol{x}_0 \sim p_{\theta,0}, \\ \boldsymbol{x}_t|\boldsymbol{x}_0 \sim q_t(\boldsymbol{x}_t|\boldsymbol{x}_0)}} \boldsymbol{u}(\boldsymbol{x}_t)^T \boldsymbol{s}_{p_{\theta,t}}(\boldsymbol{x}_t) = \mathbb{E}_{\boldsymbol{x}_t \sim p_{\theta,t}} \boldsymbol{u}(\boldsymbol{x}_t)^T \boldsymbol{s}_{p_{\theta,t}}(\boldsymbol{x}_t)$$

$$= \int p_{\theta,t}(\boldsymbol{x}_t) \boldsymbol{u}(\boldsymbol{x}_t)^T \boldsymbol{s}_{p_{\theta,t}}(\boldsymbol{x}_t) \mathrm{d}\boldsymbol{x}_t$$

$$= \int p_{\theta,t}(\boldsymbol{x}_t) \boldsymbol{u}(\boldsymbol{x}_t)^T \int \frac{\nabla_{\boldsymbol{x}_t} \log q_t(\boldsymbol{x}_t|\boldsymbol{x}_0) p_{\theta,t}(\boldsymbol{x}_0) q_t(\boldsymbol{x}_t|\boldsymbol{x}_0)}{p_{\theta,t}(\boldsymbol{x}_t)} \mathrm{d}\boldsymbol{x}_0 \mathrm{d}\boldsymbol{x}_t$$

$$= \int \boldsymbol{u}(\boldsymbol{x}_t)^T \int \nabla_{\boldsymbol{x}_t} \log q_t(\boldsymbol{x}_t|\boldsymbol{x}_0) p_{\theta,t}(\boldsymbol{x}_0) q_t(\boldsymbol{x}_t|\boldsymbol{x}_0) \mathrm{d}\boldsymbol{x}_0 \mathrm{d}\boldsymbol{x}_t$$

$$= \int \int \boldsymbol{u}(\boldsymbol{x}_t)^T \nabla_{\boldsymbol{x}_t} \log q_t(\boldsymbol{x}_t|\boldsymbol{x}_0) p_{\theta,t}(\boldsymbol{x}_0) q_t(\boldsymbol{x}_t|\boldsymbol{x}_0) \mathrm{d}\boldsymbol{x}_0 \mathrm{d}\boldsymbol{x}_t$$

$$= \mathbb{E}_{\substack{\boldsymbol{x}_0 \sim p_{\theta,0}, \\ \boldsymbol{x}_t|\boldsymbol{x}_0 \sim q_t(\boldsymbol{x}_t|\boldsymbol{x}_0)}} \boldsymbol{u}(\boldsymbol{x}_t)^T \nabla_{\boldsymbol{x}_t} \log q_t(\boldsymbol{x}_t|\boldsymbol{x}_0)$$

If we take the $\theta$ gradient on both sides of (A.5), we have

$$0 = \mathbb{E}_{\substack{\boldsymbol{x}_0 \sim p_{\theta,0}, \\ \boldsymbol{x}_t|\boldsymbol{x}_0 \sim q_t(\boldsymbol{x}_t|\boldsymbol{x}_0)}} \left\{ \frac{\partial}{\partial \boldsymbol{x}_t} \left[ \boldsymbol{u}(\boldsymbol{x}_t)^T \left\{ \boldsymbol{s}_{p_{\theta,t}}(\boldsymbol{x}_t) - \nabla_{\boldsymbol{x}_t} \log q_t(\boldsymbol{x}_t|\boldsymbol{x}_0) \right\} \right] \frac{\partial \boldsymbol{x}_t}{\partial \theta} \right.$$
$$\left. - \boldsymbol{u}(\boldsymbol{x}_t)^T \frac{\partial}{\partial \boldsymbol{x}_0} \left[ \nabla_{\boldsymbol{x}_t} \log q_t(\boldsymbol{x}_t|\boldsymbol{x}_0) \right] \frac{\partial \boldsymbol{x}_0}{\partial \theta} \right\} + \mathbb{E}_{\boldsymbol{x}_t \sim p_{\theta,t}} \boldsymbol{u}(\boldsymbol{x}_t)^T \frac{\partial}{\partial \theta} \left\{ \boldsymbol{s}_{p_{\theta,t}}(\boldsymbol{x}_t) \right\} \qquad \text{(A.6)}$$

So we have an identity

$$\mathbb{E}_{\boldsymbol{x}_t \sim p_{\theta,t}} \boldsymbol{u}(\boldsymbol{x}_t)^T \frac{\partial}{\partial \theta} \left\{ \boldsymbol{s}_{p_{\theta,t}}(\boldsymbol{x}_t) \right\} = -\frac{\partial}{\partial \theta} \mathbb{E}_{\substack{\boldsymbol{x}_0 \sim p_{\theta,0}, \\ \boldsymbol{x}_t|\boldsymbol{x}_0 \sim q_t(\boldsymbol{x}_t|\boldsymbol{x}_0)}} \left\{ \boldsymbol{u}(\boldsymbol{x}_t) \left\{ \boldsymbol{s}_{p_{\mathrm{sg}[\theta],t}}(\boldsymbol{x}_t) - \nabla_{\boldsymbol{x}_t} \log q_t(\boldsymbol{x}_t|\boldsymbol{x}_0) \right\} \right\}$$

Notice that the left-hand side of equation (A.7) can be interpreted as the gradient of the loss function when the parameter dependency of the sampling distribution is cut off, i.e.

$$\mathbb{E}_{\boldsymbol{x}_t \sim p_{\theta,t}} \boldsymbol{u}(\boldsymbol{x}_t)^T \frac{\partial}{\partial \theta} \left\{ \boldsymbol{s}_{p_{\theta,t}}(\boldsymbol{x}_t) \right\} = \frac{\partial}{\partial \theta} \mathbb{E}_{\boldsymbol{x}_t \sim p_{\mathrm{sg}[\theta],t}} \left\{ \boldsymbol{u}(\boldsymbol{x}_t)^T \boldsymbol{s}_{p_{\theta,t}}(\boldsymbol{x}_t) \right\} \qquad \text{(A.7)}$$

Therefore we have the final equation

$$\frac{\partial}{\partial \theta} \mathbb{E}_{\boldsymbol{x}_t \sim p_{\mathrm{sg}[\theta],t}} \left\{ \boldsymbol{u}(\boldsymbol{x}_t)^T \boldsymbol{s}_{p_{\theta,t}}(\boldsymbol{x}_t) \right\} = -\frac{\partial}{\partial \theta} \mathbb{E}_{\substack{\boldsymbol{x}_0 \sim p_{\theta,0}, \\ \boldsymbol{x}_t|\boldsymbol{x}_0 \sim q_t(\boldsymbol{x}_t|\boldsymbol{x}_0)}} \left\{ \boldsymbol{u}(\boldsymbol{x}_t) \left\{ \boldsymbol{s}_{p_{\mathrm{sg}[\theta],t}}(\boldsymbol{x}_t) - \nabla_{\boldsymbol{x}_t} \log q_t(\boldsymbol{x}_t|\boldsymbol{x}_0) \right\} \right\} \qquad \text{(A.8)}$$

which holds for arbitrary function $\boldsymbol{u}(\cdot)$ and parameter $\theta$. If we set

$$\boldsymbol{u}(\boldsymbol{x}_t) = \mathbf{d}'(\boldsymbol{y}_t)$$
$$\boldsymbol{y}_t = \boldsymbol{s}_{p_{\mathrm{sg}[\theta],t}}(\boldsymbol{x}_t) - \boldsymbol{s}_{q_t}(\boldsymbol{x}_t)$$

Then we formally have

$$\frac{\partial}{\partial \theta} \mathbb{E}_{\boldsymbol{x}_t \sim p_{\mathrm{sg}[\theta],t}} \left\{ \mathbf{d}'(\boldsymbol{y}_t) \right\}^T \left\{ \boldsymbol{s}_{p_{\theta,t}}(\boldsymbol{x}_t) \right\}$$

$$= \frac{\partial}{\partial \theta} \mathbb{E}_{\substack{\boldsymbol{x}_0 \sim p_{\theta,0}, \\ \boldsymbol{x}_t|\boldsymbol{x}_0 \sim q_t(\boldsymbol{x}_t|\boldsymbol{x}_0)}} \left\{ -\mathbf{d}'(\boldsymbol{y}_t) \right\}^T \left\{ \boldsymbol{s}_{p_{\theta,t}}(\boldsymbol{x}_t) - \nabla_{\boldsymbol{x}_t} \log q_t(\boldsymbol{x}_t|\boldsymbol{x}_0) \right\} \qquad \text{(A.9)}$$

This means that we can use the $\theta$ gradient of a tractable loss:

$$\mathbb{E}_{\substack{t,\boldsymbol{x}_0 \sim p_{\theta,0}, \\ \boldsymbol{x}_t|\boldsymbol{x}_0 \sim q_t(\boldsymbol{x}_t|\boldsymbol{x}_0)}} w(t)\left\{-\mathbf{d}'(\boldsymbol{y}_t)\right\}^T \left\{\boldsymbol{s}_{p_{\theta,t}}(\boldsymbol{x}_t) - \nabla_{\boldsymbol{x}_t} \log q_t(\boldsymbol{x}_t|\boldsymbol{x}_0)\right\} \tag{A.10}$$

to replace the wanted $\theta$ gradient (A.3), which can minimize the regularization loss (A.2).

Combining $r(\boldsymbol{x}_0, \boldsymbol{c})$ and (A.10), we have the practical loss

$$\mathcal{L}_{DI*}(\theta) = \mathbb{E}_{\substack{\boldsymbol{z} \sim p_z, \\ \boldsymbol{x}_0 = g_\theta(\boldsymbol{z})}} \left[ -\alpha r(\boldsymbol{x}_0, \boldsymbol{c}) \right. \tag{A.11}$$
$$\left. + \mathbb{E}_{\substack{t,\boldsymbol{x}_t|\boldsymbol{x}_0 \\ \sim q_t(\boldsymbol{x}_t|\boldsymbol{x}_0)}} w(t)\left\{-\mathbf{d}'(\boldsymbol{y}_t)\right\}^T \left\{\boldsymbol{s}_{p_{\mathrm{sg}[\theta],t}}(\boldsymbol{x}_t) - \nabla_{\boldsymbol{x}_t} \log q_t(\boldsymbol{x}_t|\boldsymbol{x}_0)\right\}\mathrm{d}t \right]$$

$\square$

**Remark A.2.** In practice, most commonly used forward diffusion processes can be expressed as a form of scale and noise addition:

$$\boldsymbol{x}_t = \alpha(t)\boldsymbol{x}_0 + \beta(t)\epsilon, \quad \epsilon \sim \mathcal{N}(\epsilon; \mathbf{0}, \mathbf{I}). \tag{A.12}$$

So the term $\boldsymbol{x}_t$ in equation (A.11) can be instantiated as $\boldsymbol{z} \sim p_z$, $\epsilon \sim \mathcal{N}(\epsilon; \mathbf{0}, \boldsymbol{I})$, $\boldsymbol{x}_t = \alpha(t)\boldsymbol{x}_0 + \beta(t)\epsilon$.

### A.2. Proof of Theorem 3.2

*Proof.* Recall the definition of the classifier-free reward (3.6). The negative reward writes

$$-r(\boldsymbol{x}_0, \boldsymbol{c}) = -\mathbb{E}_{t,\boldsymbol{x}_t \sim p_{\theta,t}} w(t) \log \frac{p_{ref}(\boldsymbol{x}_t|t, \boldsymbol{c})}{p_{ref}(\boldsymbol{x}_t|t)}$$

This reward will put a higher reward on those samples that have higher class-conditional probability than unconditional probability, therefore encouraging class-conditional sampling. It is clear that

$$\frac{\partial}{\partial \theta}\left\{-r(\boldsymbol{x}_0, \boldsymbol{c})\right\} = -\mathbb{E}_{t,\boldsymbol{x}_t \sim p_{\theta,t}} w(t)\left\{\nabla_{\boldsymbol{x}_t} \log p_{ref}(\boldsymbol{x}_t|t, \boldsymbol{c}) - \nabla_{\boldsymbol{x}_t} \log p_{ref}(\boldsymbol{x}_t|t)\right\}\frac{\partial \boldsymbol{x}_t}{\partial \theta}$$
$$= -\mathbb{E}_{t,\boldsymbol{x}_t \sim p_{\theta,t}} w(t)\left\{\boldsymbol{s}_{ref}(\mathrm{sg}[\boldsymbol{x}_t]|t, \boldsymbol{c}) - \boldsymbol{s}_{ref}(\mathrm{sg}[\boldsymbol{x}_t]|t, \varnothing)\right\}\frac{\partial \boldsymbol{x}_t}{\partial \theta} \tag{A.13}$$

Therefore, we can see that the equivalent loss

$$\mathcal{L}_{cfg}(\theta) = \mathbb{E}_{\substack{t,\boldsymbol{z} \sim p_z, \boldsymbol{x}_0 = g_\theta(\boldsymbol{z}|\boldsymbol{c}) \\ \boldsymbol{x}_t|\boldsymbol{x}_0 \sim p(\boldsymbol{x}_t|\boldsymbol{x}_0)}} w(t)\left\{\boldsymbol{s}_{ref}(\mathrm{sg}[\boldsymbol{x}_t]|t, \boldsymbol{c}) - \boldsymbol{s}_{ref}(\mathrm{sg}[\boldsymbol{x}_t]|t, \varnothing)\right\}^T \boldsymbol{x}_t \tag{A.14}$$

recovers the gradient formula (A.13). $\square$

## B. Additional Discussions

### B.1. Meanings of Hyper-parameters.

**Meanings of Hyper-parameters.** As in Algorithm 1, the overall algorithms consist of two alternative updating steps. The first step is to update $\psi$ of the assistant diffusion model by fine-tuning it with student-generated data. Therefore the assistant diffusion $\boldsymbol{s}_\psi(\boldsymbol{x}_t|t, \boldsymbol{c})$ can approximate the score function of student one-step model distribution. This step means that the assistant diffusion needs to communicate with the student to know the student's status. The second step updates the one-step by minimizing the tractable loss (C.1) using SGD-based optimization algorithms such as Adam (Kingma & Ba, 2014). This step means that the teacher and the assistant diffusion discuss and incorporate the student's interests to instruct the student one-step model.

As we can see in Algorithm 1 (as well as Algorithm 2). Each hyperparameter has its intuitive meaning. The reward scale parameter $\alpha_{rew}$ controls the strength of human preference alignment. The larger the $\alpha_{rew}$ is, the stronger the one-step

model is aligned with human preferences. However, the drawback for a too large $\alpha_{rew}$ might be the loss of diversity and reality. Besides, we empirically find that larger $\alpha_{rew}$ leads to richer generation details and better generation layouts. But a very large $\alpha_{rew}$ results in unrealistic and painting-like images.

The CFG reward scale controls the strength of using CFG rewards when training. We empirically find that the best CFG scale for Diff-Instruct* is the same as the best CFG scale for sampling from the reference diffusion model. However, $\alpha_{cfg}$ may conflicts with $\alpha_{rew}$. In the Stable Diffusion 1.5 experiment, we find that using a large CFG reward scale leads to worse human preferences. Therefore, the proper combination of $(\alpha_{rew}, \alpha_{cfg})$ asks for careful tuning.

The diffusion model weighting $\lambda(t)$ and the one-step model loss weighting $w(t)$ controls the strengths put on each time level of updating assistant diffusion and the student generator. We empirically find that it is decent to set $\lambda(t)$ to be the same as the default training weighting function for the reference diffusion. And it is decent to set the $w(t) = 1$ for all time levels in practice. In the following section, we give more discussions on Diff-Instruct*.

### B.2. Discussions on Diff-Instruct*

**Flexible Choices of Divergences.** Various choices of distance function $\mathbf{d}(.)$ result in different training algorithms. In this part, we discuss two instances. The first choice distance function is a simple squared distance, i.e. $\mathbf{d}(\boldsymbol{y}_t) = \|\boldsymbol{y}_t\|_2^2$. The corresponding derivative term writes $\mathbf{d}'(\boldsymbol{y}_t) = 2\boldsymbol{y}_t$. In fact, such a distance function recovers the practical diffusion distillation loss studied in Zhou et al. (2024b;a). The second distance is the pseudo-Huber distance, which shows more robust performances than the simple squared distance. The pseudo-Huber distance is defined with $\boldsymbol{d}(\boldsymbol{y}) := \sqrt{\|\boldsymbol{y}_t\|_2^2 + c^2} - c$, where $c$ is a pre-defined positive constant. The corresponding regularization loss (3.4) writes

$$\mathbf{D}^{[0,T]}(p_\theta, p_{ref}) = -\left\{\frac{\boldsymbol{y}_t}{\sqrt{\|\boldsymbol{y}_t\|_2^2 + c^2}}\right\}^T \left\{\boldsymbol{s}_\psi(\boldsymbol{x}_t, t) - \nabla_{\boldsymbol{x}_t} \log q_t(\boldsymbol{x}_t|\boldsymbol{x}_0)\right\}. \tag{B.1}$$

Here $\boldsymbol{y}_t := \boldsymbol{s}_{p_{\mathrm{sg}[\theta],t}}(\boldsymbol{x}_t) - \boldsymbol{s}_{q_t}(\boldsymbol{x}_t)$.

**DI\* Does Not Need Image Data for Training.** One appealing advantage of DI* is the image data-free property, which means that DI* requires neither the image datasets nor synthetic images that are generated by reference diffusion models. This strength distinguishes DI* from previous fine-tuning methods such as generative adversarial training (Goodfellow et al., 2014) which require training additional neural classifiers over image data, as well as those fine-tuning methods over large-scale synthetic or curated datasets.

## C. Qualitative Results

**Evaluation Details** To quantitatively evaluate the performances of the one-step model trained with different alignment settings, we compare our generators elaborately with other $1024 \times 1024$ resolution open-sourced models that are either based on SDXL diffusion models or larger models such as FLUX-dev and SD 3.5 large. For all models, we compute four standard scores: the Image Reward (Xu et al., 2023), the Aesthetic Score (Schuhmann, 2022), the PickScore(Kirstain et al., 2023), and the CLIP score(Radford et al., 2021). Since most existing literature tests the human preference scores with different prompts which are possibly not available, to make a fair comparison, in our experiment, we use 30k prompts from the COCO-2014 (Lin et al., 2014) validation dataset and intensively evaluate a wide range of existing open-sourced models. All models are tested with the same prompts and the same computing devices.

Besides the COCO prompts, we also evaluate one-step models with open-sourced models with Human Preference Score v2.0 (HPSv2.0) (Wu et al., 2023) over their benchmark prompts. The HPS is a widely used standard benchmark that evaluates models' capability of generating images of 4 styles: Animation, concept art, Painting, and Photo. The score reflects the prompt following and the human preference strength of text-to-image models. We use the HPSv2's [1] default protocols for evaluations.

---

[1] https://github.com/tgxs002/HPSv2

**Algorithm 2:** Diff-Instruct*.

**Input:** prompt dataset $\mathcal{C}$, one-step model $g_\theta(\boldsymbol{x}_0|\boldsymbol{z}, \boldsymbol{c})$, prior distribution $p_z$, reward model $r(\boldsymbol{x}, \boldsymbol{c})$, reward model scale $\alpha_{rew}$, CFG reward scale $\alpha_{cfg}$, reference diffusion model $\boldsymbol{s}_{ref}(\boldsymbol{x}_t|c, \boldsymbol{c})$, assistant diffusion $\boldsymbol{s}_\psi(\boldsymbol{x}_t|t, \boldsymbol{c})$, forward diffusion $p_t(\boldsymbol{x}_t|\boldsymbol{x}_0)$ (2.1), assistant diffusion updates rounds $K_{TA}$, time distribution $\pi(t)$, diffusion model weighting $\lambda(t)$, student loss time weighting $w(t)$.

**while** *not converge* **do**

    freeze $\theta$, update $\psi$ for $K_{TA}$ rounds using SGD by minimizing

$$\mathcal{L}(\psi) = \mathbb{E}_{\substack{\boldsymbol{c}\sim\mathcal{C}, \boldsymbol{z}\sim p_z, t\sim\pi(t) \\ \boldsymbol{x}_0 = g_\theta(\boldsymbol{z}|\boldsymbol{c}), \boldsymbol{x}_t|\boldsymbol{x}_0\sim p_t(\boldsymbol{x}_t|\boldsymbol{x}_0)}} \lambda(t)\|\boldsymbol{s}_\psi(\boldsymbol{x}_t|t, \boldsymbol{c}) - \nabla_{\boldsymbol{x}_t}\log p_t(\boldsymbol{x}_t|\boldsymbol{x}_0)\|_2^2 \mathrm{d}t.$$

    freeze $\psi$, update $\theta$ using SGD by minimizing loss

$$\mathcal{L}_{DI*}(\theta) = \mathbb{E}_{\substack{\boldsymbol{c}\sim\mathcal{C}, \boldsymbol{z}\sim p_z, \\ \boldsymbol{x}_0 = g_\theta(\boldsymbol{z}, \boldsymbol{c})}} \Bigg\{ -\alpha_{rew}\cdot r(\boldsymbol{x}_0, \boldsymbol{c}) + \mathbb{E}_{\substack{t\sim\pi(t), \\ \boldsymbol{x}_t|\boldsymbol{x}_0\sim p_t(\boldsymbol{x}_t|\boldsymbol{x}_0)}} \Bigg[$$
$$-w(t)\big\{\mathbf{d}'(\boldsymbol{s}_\psi(\boldsymbol{x}_t|t, \boldsymbol{c}) - \boldsymbol{s}_{ref}(\boldsymbol{x}_t|t, \boldsymbol{c}))\big\}^T \big\{\boldsymbol{s}_\psi(\boldsymbol{x}_t|t, \boldsymbol{c}) - \nabla_{\boldsymbol{x}_t}\log p_t(\boldsymbol{x}_t|\boldsymbol{x}_0)\big\}$$
$$+ \alpha_{cfg}\cdot w(t)\big\{\boldsymbol{s}_{ref}(\mathrm{sg}[\boldsymbol{x}_t]|t, \boldsymbol{c}) - \boldsymbol{s}_{ref}(\mathrm{sg}[\boldsymbol{x}_t]|t, \boldsymbol{\varnothing})\big\}^T \boldsymbol{x}_t\Bigg]\Bigg\} \quad\quad\text{(C.1)}$$

**end**
**return** $\theta, \psi$.

## C.1. Pytorch style pseudo-code of Score Implicit Matching

In this section, we provide a PyTorch-style pseudo-code for the algorithm below.

```python
import torch
import torch.nn as nn
import torch.optim as optim
import copy

use_cfg = True
use_reward = True

# Initialize generator G
G = Generator()

## load teacher DM
Drf = DiffusionModel().load('/path_to_ckpt').eval().requires_grad_(False)
Dta = copy.deepcopy(Drf) ## initialize online DM with teacher DM
r = RewardModel() if use_reward else None

# Define optimizers
opt_G = optim.Adam(G.parameters(), lr=0.001, betas=(0.0, 0.999))
opt_Sta = optim.Adam(Dta.parameters(), lr=0.001, betas=(0.0, 0.999))

# Training loop
while True:
    ## update Dta
    Dta.train().requires_grad_(True)
    G.eval().requires_grad_(False)

    ## update assistant diffusion
    prompt = batch['prompt']
    z = torch.randn((1024, 4, 64, 64), device=G.device)
    with torch.no_grad():
        fake_x0 = G(z,prompt)

    sigma = torch.exp(2.0*torch.randn([1,1,1,1], device=fake_x0.device) - 2.0)
    noise = torch.randn_like(fake_x0)
```

```
fake_xt = fake_x0 + sigma*noise
pred_x0 = Dta(fake_xt, sigma, prompt)

weight = compute_diffusion_weight(sigma)

batch_loss = weight * (pred_x0 - fake_x0)**2
batch_loss = batch_loss.sum([1,2,3]).mean()

optimizer_Dta.zero_grad()
batch_loss.backward()
optimizer_Dta.step()

## update G
Dta.eval().requires_grad_(False)
G.train().requires_grad_(True)

prompt = batch['prompt']
z = torch.randn((1024, 4, 64, 64), device=G.device)
fake_x0 = G(z, prompt)

sigma = torch.exp(2.0*torch.randn([1,1,1,1], device=fake_x0.device) - 2.0)
noise = torch.randn_like(fake_x0)
fake_xt = fake_x0 + sigma*noise

with torch.no_grad():
    if use_cfg:
        cfg_vector = (Drf(fake_xt, sigma, prompt) - Drf(fake_xt, sigma, None)
    else:
        cfg_vector = None

    pred_x0_rf = Drf(fake_xt, sigma, prompt)
    pred_x0_ta = Dta(fake_xt, sigma, prompt)

denoise_diff = pred_x0_ta - pred_x0_rf
adp_wgt = torch.sqrt(denoise_diff.square().sum([1,2,3], keepdims=True) + phuber_c**2)
weight = compute_G_weight(sigma, denoise_diff)

# compute score regularization loss
batch_loss = weight * denoise_diff * (fake_D_yn - D_yn)/adp_wgt

# compute explicit reward loss if needed
if use_reward:
    reward_loss = -reward_scale * r(fake_x0, prompt)
    batch_loss += reward_loss

# compute cfg reward loss if needed
if use_cfg:
    cfg_reward_loss = cfg_scale * cfg_vector*fake_x0
    batch_loss += cfg_reward_loss

batch_loss = batch_loss.sum([1,2,3]).mean()

optimizer_G.zero_grad()
batch_loss.backward()
optimizer_G.step()
```

Listing 1: Pytorch Style Pseudo-code of Diff-Instruct*

