# OpenReview forum: "David and Goliath: Small One-step Model Beats Large Diffusion with Score Post-training"
_ICML.cc/2025/Conference — ICML 2025 poster_

### Official Review · Reviewer_zH6y · 2025-03-13

**Overall Recommendation:** 4

**Summary:**

The paper presents a novel online method for alignment fine-tuning of a one-step diffusion-based text-to-image generation model.

**Claims And Evidence:**

* The paper claims to require no data in the abstract. But in reality, it does use a dataset of prompts. I think this is misleading.
* The paper abstract seems to suggest the paper trains a one-step generation model and then fine-tunes it for preferences. But in practice, the authors start from a pre-trained one-step model. It could be made clear.

**Essential References Not Discussed:**

MaPO [1] in the preference alignment literature.

Imagine-Flash [2], Hyper-SD [3] in the one-step model literature.

**Experimental Designs Or Analyses:**

* How general is this method? Could the authors apply it to a smaller model like PixArt-Alpha [1] and demonstrate similar results?
* What are the memory requirements of this method? The method seems to need three models in memory, which leads me to worry about its memory-intensive nature.
* Could the method be implemented with LoRA [2]? For example, in C.1,  `Drf` could be the base reference model and `Dta` could be injected with the LoRA layers. Then during forwarding with `Drf`, the LoRA layers could be disabled and enabled when forwarding with `Dta`. If this works well, this would substantially reduce the memory requirements. An implementation of this is available in [3].
* What is `G` in C.1.? What is its architecture?
* Can DI* be applied to non-1-step models?

References

[1] PixArt-α: Fast Training of Diffusion Transformer for Photorealistic Text-to-Image Synthesis; Chen et al.; 2023; https://arxiv.org/abs/2310.00426.

[2] LoRA: Low-Rank Adaptation of Large Language Models; Hu et al.; 2021; https://arxiv.org/abs/2106.09685.

[3] StackLLaMA: A hands-on guide to train LLaMA with RLHF; Beeching et al.; 2023; https://huggingface.co/blog/stackllama.

**Methods And Evaluation Criteria:**

Proposed methods make sense. I have several points to make here.

* The proposed method is shown to improve metrics like ClipScore, PickScore, etc. However, did the authors obtain other important metrics like GenEval [1], T2I-Compbench [2]? This could provide even better insights into understanding if the method disturbs the spatial capabilities of the pre-trained model.
* It is not clear how this method performs on complex prompts. One drawback of using SDXL as the base architecture could be that it uses CLIP which restricts the prompt length only at 77. This is quite small compared to other more recent models like Flux.
* The authors use a reference model to enforce regularization during he preference alignment step. What happens when there is a reference mismatch as investigated in MaPO [3]?
* It would be better if the authors could distinguish between Diff-Instruct and Diff-Instruct* as the proposed method seems to draw a lot of parallels from Diff-Instruct.
* Are there any criteria the assistant model needs to satisfy?
* The paper has some insights on the different loss scales. I am interested to know if changing one aspect of the final loss function leads to any interesting properties (for example, does increasing the CFG influence as an implicit reward help with anything?).
*  Did the authors explore other implicit rewards and explicit rewards in a controlled setup? I think examining different external rewards could be beneficial to understand the general trends when choosing one.
* Instead of using prompts from the COCO dataset during evaluation, using the test sets of benchmark datasets like PartiPrompt [4], DrawBench [5], HPSv2-test set [6] could also be beneficial.

References

[1] GenEval: An Object-Focused Framework for Evaluating Text-to-Image Alignment; Ghosh et al.; 2023; https://arxiv.org/abs/2310.11513.

[2] T2I-CompBench: A Comprehensive Benchmark for Open-world Compositional Text-to-image Generation; Huang et al.; 2023; https://arxiv.org/abs/2307.06350.

[3] Margin-aware Preference Optimization for Aligning Diffusion Models without Reference; Hong et al.; 2024; https://arxiv.org/abs/2406.06424.

[4] Scaling Autoregressive Models for Content-Rich Text-to-Image Generation; Yu et al.; 2022; https://arxiv.org/abs/2206.10789.

[5] Photorealistic Text-to-Image Diffusion Models with Deep Language Understanding; Saharia et al.; 2022; https://arxiv.org/abs/2205.11487.

[6] Human Preference Score v2: A Solid Benchmark for Evaluating Human Preferences of Text-to-Image Synthesis; Wu et al.; 2023; https://arxiv.org/abs/2306.09341.

**Other Comments Or Suggestions:**

* Why is the blue color present in abundance?
* Why does the first equation start from 2.1.?
* In Figure 1, the second sentence seems redundant.
* In the introduction section, advancements and contributions seem to have a very redundant overlap. Consider revisiting.
* It is not clear in what is $p_\theta$ and how it is different from $g_\theta$.
* $r$ is undefined in the context of equation 3.1.
* What is $\mathbf{d}^{\prime}\left(\boldsymbol{y}_t\right)$ in equation 3.4?
* The exact checkpoints used for the models could be provided as footnotes.
* The training related details are missing.

**Other Strengths And Weaknesses:**

I have covered them in the other sections.

**Questions For Authors:**

Have noted in other sections.

**Relation To Broader Scientific Literature:**

Having means to generate images speedily that follow human preferences better is beneficial.

**Theoretical Claims:**

Theorem 3.1 has been substantiated in the supplementary.

---

> ### Author Rebuttal · Authors · 2025-04-01
>
> Dear reviewer, we are delighted that you like the novelty of DIstar for the one-step diffusion model post-training. We appreciate your valuable suggestions. In the following paragraphs, we will address your concerns one by one.
>
> **Q1**. Clarifications on image-data-free property and post-training setups.
>
> **A1**. We will polish the setup as image-data free and post-training of one-step diffusion models.
>
> **Q2**. (1) Broader evaluations, such as evaluations on challenging prompts like Parti and HPSv-benchs. (2) How does DIstar perform on models with the ability to take complex prompts, such as Pixart-alpha models? (3) Explorations on other rewards.
>
> **A2**. Thanks for your useful feedback. **In Table 1 and Table 2** in our paper, we have compared DIstar with other models on Parti prompts and HPSv2.1 benchmark. In the rebuttal period, we conduct two new experiments: **(1)** DIstar with SDXL-DPO (a pre-aligned SDXL model) as the reference diffusion; **(2)** DIstars with Pixel-art-alpha model that uses ImageReward as explicit reward models. We present more quantitative results in **Table 2** (in rebuttal to **Reviewer NM9A**), and **Table 1**.
>
> **Table 1.** Quantitative comparisons of one-step SD1.5 model and one-step Pixel-art-alpha (**PAA**) models and others in Preference Score on **COCO-2017-validation** prompt dataset.
> | Model                               | Steps | Type | Params | **Image Reward** | **Aes Score** | **Pick Score** | **CLIP Score** | **HPSv2.0** |
> |--|--|--|--|--|--|--|--|--|
> | PAA-512                     | 25    | DiT  | 0.6B   | 0.82         | 6.01      | **0.227**      | **31.20**      | 28.25|
> | PAA-DI++      | 1     | DiT  | 0.6B   | 1.24     | 6.19      | 0.225      | 30.80      | 28.48 |
> | **PAA-DIstar**      | 1     | DiT  | 0.6B   | **1.31**     | **6.30**      | 0.225      | 30.84      | **28.70** |
>
> The Pixel-art-alpha (PAA) diffusion model uses the T5 text encoder, which has a much longer context length. Besides, we also explore the use of ImageReward to see if DIstar is consistent across different reward models. So, we compare DIstar on the PAA model with other open-sourced models. As **Table 1** shows, the PAA-DIstar model achieves a leading ImageReward, Aesthetic score, and HPSv2.0 score. It goes on par with the best PickScore as PAA-diffusion-25step shows.
>
> **Q3**. Discussion on the mismatch setups introduced in MaPO.
>
> **A3**. Thanks for the valuable comment. After a careful reading of the MaPO paper, we find it technically very interesting. MaPO introduced a novel approach to post-train diffusion models in cases of preference mismatch. We acknowledge that in our paper, we did not consider cases of preference mismatches, which is potentially important to study in our future. However, the DIstar and MaPO have essentially different targets in
>
> (1) MaPO targets in diffusion models, while DIstar targets in post-training of one-step models;
>
> (2) MaPO assumes access to the preference dataset, while DIstar assumes we have high-quality and diverse reward models;
>
> However, we really appreciate the solid contributions of MaPO. We will add an additional discussion on MaPO in our revision. We are glad to explore DIstar in cases of preference mismatches introduced by MaPO.
>
> **Q4**. Discuss DIstar with other related approaches.
>
> **A4**. Please see **A1** in rebuttal to **Reviewer KEDK**.
>
> **Q5**. Are there any criteria the assistant model needs to satisfy?
>
> **A5**. In general, arbitrary diffusion models can serve as an assistant model. But in practice, we initialize the assistant diffusion model with the preference diffusion model and find it works well.
>
> **Q6**. Clarify more on three losses.
>
> **A6**. Please see **A3** in the rebuttal to **Reviewer NM9A**.
>
> **Q7**. What are the memory requirements of this method? Is DIstar compatible with LoRA?
>
> **A7**. We acknowledge that DIstar needs an additional assistant diffusion model, which brings more memory costs. However, in practice, we find that the additional cost is acceptable if we use Pytorch techniques like BF16 training, Distributed data-parallel (DDP), and gradient accumulations. We did not use LoRA in order to get the best performances.
>
> **Q8**. Criteria for the assistant model. DIstar for few-step models?
>
> **A8**.  In general, arbitrary diffusion models can serve as an assistant model. DIstar for a few steps, possibly in our future work.
>
>
> **Q9**. Other comments on paper writing.
>
> **A9**. We will remove the blue colors. $p_\theta$ represents the distribution of the one-step model. $d'(y)$ in equation 3.4 represents the derivative of distance $d(y)$. We appreciate your valuable comments on the paper writing, and we will revise the paper writing in the revision. **We will add experiment details in our revision.**
>
> **We appreciate your valuable review. We sincerely hope our response has resolved your concerns. If you still have any concerns, please let us know.**

---

> > ### Comment · Reviewer_zH6y · 2025-04-02
> >
> > Thanks for the additional clarification. I will also increase my score. Good luck.

---

> > > ### Author Response · Authors · 2025-04-02
> > >
> > > Dear **Reviewer zH6y**,
> > >
> > >
> > > We are glad that we have resolved your concerns. We greatly appreciate your valuable suggestions and will incorporate them into our revision.
> > >
> > > **Authors of the submission # 7929**

---

### Official Review · Reviewer_NM9A · 2025-03-14

**Overall Recommendation:** 3

**Summary:**

This paper proposes Diff-Instruct*, a post-training method to align one-step text-to-image generative models with human preferences. The work is an evolution of “Diff-Instruct” and “Diff-Instruct++,” replacing the KL-based divergence (as in standard PPO) to a score-based divergence for regularization. Empirical results show superior performance compared to KL-based method.

## Update after rebuttal
The authors' rebuttal has addressed my concerns regarding the motivation for replacing KL-divergence and the necessity of aligning a distilled one-step diffusion model. I am raising my score to weak accept.

**Claims And Evidence:**

The claim of KL-divergence being "notorious for mode-seeking behavior" is not supported by solid experimental or theoretical evidence.
While building upon Diff-Instruct++, there could be more insights into why aligning a distilled one-step diffusion model is more effective than distilling from a pre-aligned multi-step diffusion model. Further clarification is needed to understand whether aligning a one-step model is easier and offers better performance.

**Essential References Not Discussed:**

No.

**Experimental Designs Or Analyses:**

1. The comparison with SDXL-DPO is not entirely fair, particularly regarding inference time improvements. The evaluation should also compare with a DMD-distilled version of SDXL-DPO.

2. The ablation study requires further detail; e.g., clarify the trade-offs between the three loss signals (main human reward, CFG reward, and regularization loss).

**Methods And Evaluation Criteria:**

The idea of using score-based divergence to regularize one-step diffusion model is novel. The design choices and discussion of incorporating CFG are interesting. The derivation  of the proposed method is reasonable and clear.

The evaluation criteria appear appropriate.

**Other Comments Or Suggestions:**

1. In Fig. 3 (supplementary), please clarify what “TA” stands for.
2. In Algorithm 1, although the notation “sg” appears earlier, it should still be clearly defined so that the algorithm is fully self-contained.

**Other Strengths And Weaknesses:**

Strengths

The idea of using score-based divergence to regularize one-step diffusion model is novel. The design choices and discussion of incorporating CFG are interesting. The derivation  of the proposed method is reasonable and clear. Empirical results show superior performance compared to KL-based method.


Weaknesses

The paper could better study the trade-off between the three loss signals (human reward, CFG reward, and regularization). For instance, if the optimization directions of these components conflict, what is the impact on text-image alignment, human preference, and image quality?

**Questions For Authors:**

1. Why is aligning a distilled one-step diffusion model better than distilling from a pre-aligned multi-step diffusion model? Is aligning a one-step model inherently easier, and what evidence or insight supports this claim?

2. If the optimization directions of the three loss signals (human reward, CFG reward, and regularization) conflict, what is the impact on text-image alignment, human preference, and image quality?

**Relation To Broader Scientific Literature:**

The paper is an evolution of “Diff-Instruct++” (Luo 2024). The paper also builds upon one-step diffusion models like DMD-v2 (Yin et al., 2024) and draw inspiration from score-based divergences from 1-step diffusion distillation (Luo et al., 2024c; Zhou et al., 2024b).

**Theoretical Claims:**

The theoretical derivation seems plausible, but more insight into the core motivation (why score-based divergence is better than KL-divergence) would strengthen the work.

Minor issues: The paper should be more self-contained, explicitly stating new conclusions (e.g., “As first pointed out by Luo (2024), classifier-free guidance is related to an implicit reward function”) for readers less familiar with the background.

---

> ### Author Rebuttal · Authors · 2025-04-01
>
> Dear reviewer, we are glad that you like our novelty of regularizing the one-step diffusion model during post-training. In the following paragraphs, we will answer your questions one by one.
>
> **Q1**. Experimental or theoretical evidence that shows mode-collapse issues of one-step diffusion distilling using KL-divergences.
>
> **A1**. We appreciate your good question! We will answer your question from both theoretical and empirical perspectives.
>
> **Theoretical perspective**: If we use $p_ r$ to represent reference distribution and $p_{g}$ the one-step generator distribution. The KL divergence between $p_ g$ and $p_ r$ is defined as:
> $D_ {KL}(p_ g||p_ r) = \mathbb{E}_ {x\sim p_g}\log \frac{p_ g (x)}{p_ r (x)}$
> while the general score-based divergence with a form $\mathcal{D}_ {SD}(p_ g,p_ d) = \mathbb{E}_ {x\sim \pi} d(\nabla_ x \log p_ g (x) - \nabla_ x \log p_ d (x))$.
> KL divergence is long believed to be bothered by mode-collapse issues because the likelihood ratio $\frac{p_ g (x)}{p_ r (x)}$ will be ill-defined if the  $p_ g$ and $p_ d$ have misaligned density support. However, the score-based divergence does not have such a "ratio" that will be ill-posed and, therefore, is safe in the case when $p_ g$ and $p_ r$ have misaligned support. Besides, the distance function $d()$ of general score-based divergence also brings flexibility in defining divergences to properly measure the difference between two distributions in a robust magnitude. For instance, our used Pseudo-Huber distance (line 259)  has a self-normalization form, which also helps stabilize the gradient of score-divergence to a unit scale, potentially further helping address the mode-collapse issue.
>
> **Empirical perspective**: We conduct a new experiment on one-step diffusion distillation on the CIFAR10 unconditional. The FID and Recall of KL and score-divergence model are (3.70,0.52) and (2.01,0.60). This shows that distilling using KL divergence leads to both worse FID and Recall.
>
> **Q2**. Is aligning a distilled one-step diffusion model more effective than distilling from a pre-aligned multi-step diffusion model? If so or not so, why or why not?
>
> **A2**. This is a very good question! We will answer via three perspectives:
>
> + First of all, distillation takes far more than compute than post-training: roughly 8:1 GPU hours. Therefore, in this paper, we focus on post-training that improves existing models with minimal costs.
>
> + Second, **we appreciate your good intuition** and find that if we **replace the naive reference diffusion with an aligned diffusion** (like SDXL-DPO) in DI-star, we can even accelerate the post-training process with better final results. In the rebuttal phase, **we conduct a new experiment** (in **Table 2**) that compares DI-star w/o pre-aligned reference diffusion, as well as other DMD2-like models.
>
> **Experiment Setup.** We replace SDXL with SDXL-DPO as the reference diffusion model. Then, we compare models after post-training using DI-star, DMD, and DI++.
>
> As **Table 2** shows, **in DI-star (Score-based PPO), both reward models and pre-aligned reference diffusion contribute to the performance improvements of one-step models in post-training**. Besides, we find that DI++(KL-PPO) will harm the contribution of the reward model and pre-aligned diffusion.
>
> **Table 2.** Quantitative comparisons of Preference Scores after post-training using SDXL-DPO on **Parti prompts (left part)** and **HPSv2.1 score (right part)**.
> | Model                               | Steps | Type | Params | Image Reward | Aes Score | Pick Score | CLIP Score | HPSv2.0 | HPSv2.1 |
> |--|--|--|--|--|--|--|--|--|--|
> | SDXL-DPO     | 50    | UNet  | 2.6B   | 1.102     | 5.77      | 0.2290      | 33.03      | | 30.42 |
> | DIstar-SDXL-DPO    | 1    | UNet  | 2.6B   | 1.160     |  5.84      | 0.2324      | 32.85      | 28.53 | 31.39 |
> | SIM-SDXL-DPO     | 1    | UNet  | 2.6B   | 1.063    |  5.79      | 0.2270      | 32.83      | 28.42 | 30.49|
> | DI++-SDXL-DPO     | 1    | UNet  | 2.6B   | 0.897     |  5.41      | 0.2225      | 33.06     | 27.86 | 28.07 |
> | DMD-SDXL-DPO     | 1    | UNet  | 2.6B   | 0.974     |  5.57      | 0.2244      | 33.07      | 28.15 | 29.29 |
> | DIstar-SDXL-DPO (Long, Best)    | 1    | UNet  | 2.6B   |  **1.210**    |   **5.90**     |   **0.2342**    |   32.91    | **28.81** | **32.25** |
>
> **Q3**. More clarifications of the trade-offs between the three loss signals
>
> **A3**.
>
> + We find that CFG-implicit reward easily causes over-saturated colors. On the contrary, CFG-enhanced reference diffusion will not cause issues like over-saturated colors.
>
> + ImageReward is conflict to CFG-rewards and CFG-enhanced divergences. However, PickScore is not conflicted with CFG-enhanced divergence loss, which makes both losses improve the one-step model during post-training.
>
> **We appreciate your valuable review. We sincerely hope our response has resolved your concerns. If you still have any concerns, please let us know.**

---

> > ### Comment · Reviewer_NM9A · 2025-04-07
> >
> > Thank you for the author's response. The rebuttal has addressed most of my concerns. However, in the theoretical explanation of the motivation, it would be clearer if the author could elaborate on what is meant by "misaligned density support" and explain why the mode-collapse issues associated with KL-divergence are related to this concept.

---

> > > ### Author Response · Authors · 2025-04-07
> > >
> > > **Dear Reviewer NM9A**,
> > >
> > > We are glad that we have addressed most of your concerns. Thanks for your engagement in the discussion.
> > >
> > > **Q1.** It would be clearer if the author could elaborate on what is meant by "misaligned density support" and explain why the mode-collapse issues associated with KL-divergence are related to this concept.
> > >
> > > **A1.** We are sorry for the confusion. In our rebuttal, the **"misaligned density support"** means the cases when two distributions $p_g$ and $p_r$ do not have the same density support. For instance, if $p_g$ is a standard Gaussian distribution $p_g=\mathcal{N}(0,I)$, while $p_r$ is a uniform distribution defined in the unit cube in $\mathcal{R}^D$. The $p_g$ is supported (has legal density) on whole $\mathcal{R}^D$ space, while $p_r$ only has legal density in the unit cube
> > > $\\{ x \in \mathbb{R}^D : 0 \leq x_i \leq 1, \forall i = 1, \dots, D \\}$,
> > >  and zero density outside the unit cube.
> > >
> > > **So why does KL divergence potentially result in mode collapse in cases of misaligned density?**
> > >
> > > We follow the same notation in the above paragraph. Now, for each point $x$ outside of the unit cube, the $p_g(x)$ has a finite value, while $p_r(x)$ is zero. Therefore, the KL divergence between $p_g$ and $p_r$ will be ill-posed (or ill-defined as the infinite), $\mathcal{D}_ {KL}(p_g||p_r)=\int_{\mathcal{R}^D} \log \frac{p_g(x)}{p_r(x)} dx =+\infty$. This means that any gradient-based optimization algorithm can not minimize such a KL divergence to let $p_g$ to be distributed as $p_r$ because the $\infty$ KL divergence can not provide any useful gradient for optimization. As a comparison, the score-based divergence is not defined through a "dangerous density ratio" and, therefore, is possibly more robust to mode-collapse issues caused by misaligned supports.
> > >
> > > In the above paragraphs, we give an intuitive understanding of why KL divergence will potentially lead to mode collapse in the case of misaligned supports. However, we do acknowledge that more explorations with strict theoretical arguments will be very cool in future work.
> > >
> > > We sincerely appreciate your great intuition and constructive comments, which we will incorporate in our revision. **If you still have any concerns, please let us know, and we are glad to provide more clarifications.**
> > >
> > > **Authors of the submission #7929**

---

### Official Review · Reviewer_KEDK · 2025-03-17

**Overall Recommendation:** 3

**Summary:**

The paper proposes Diff-Instruct* (DI*), a new post-training framework for one-step text-to-image generative models, aiming to align their outputs with human preferences without requiring image data. The method leverages score-based reinforcement learning from human feedback (RLHF), optimizing a human reward function while maintaining closeness to a reference diffusion model. Instead of conventional KL divergence for regularization, the authors introduce a score-based divergence, which is theoretically justified and empirically shown to improve model performance. The authors trained a one-step generator trained via Diff-Instruct*, called DI*-SDXL-1step model, capable of generating 1024×1024 images in a single step. It outperforms the 12B FLUX-dev model on key human preference metrics while running much faster.

**Claims And Evidence:**

The paper claims that score-based divergence regularization is superior to KL-divergence for RLHF in generative models, which is supported by the ablation studies shown in Table 2. However, in Lines 375–378, the authors refer to Figure 1 for visualizations of DI++ with KL divergence to support their claim that DI++ tends to collapse into painting-like images with oversaturated colors and lighting, which lack diversity despite high rewards. However, Figure 1 only displays examples of DI*.

**Essential References Not Discussed:**

N/A

**Experimental Designs Or Analyses:**

Ablation studies are comprehensive and quantitative results are strong and convincing, with DI* showing consistent improvements across multiple reward configurations. It would be better if the authors could present more qualitative examples comparing to KL-based models, such as DI++-SDXL, to verify the motivation of the method.

**Methods And Evaluation Criteria:**

The proposed method is reasonable, and the evaluation experiments on SDXL using the Parti prompt benchmark, the COCO-2014-30K benchmark, and Human preference scores are also well-justified.

**Other Comments Or Suggestions:**

See Claims And Evidence.

**Other Strengths And Weaknesses:**

Strengths:

- The proposed method demonstrates performance improvements over competing approaches across diverse benchmarks, supported by both quantitative and qualitative evaluations.

- The paper provides thorough ablation studies, which effectively isolate and quantify the contribution of each component to the observed performance gains, offering strong empirical evidence for the method's design.

Weaknesses:

- The application of score-based distribution matching, while effective, lacks significant novelty, as it has been previously explored in papers like DMD.

- While quantitative results suggest the superiority of score-based PPO over KL-based methods, the qualitative analysis requires expansion. Specifically, more illustrative examples are needed to substantiate the claim of improved density preservation.

**Questions For Authors:**

See Strengths And Weaknesses.

**Relation To Broader Scientific Literature:**

The paper falls within the area of diffusion distillation and RLHF for diffusion models. The related works are generally well-cited.

**Theoretical Claims:**

The paper introduces a tractable reformulation of score-based divergence for RLHF. The derivation is plausible and well-structured, but some steps could benefit from more intuition from a theoretical perspective, such as why the score-based divergence has a better diversity-preserving property than KL.

---

> ### Author Rebuttal · Authors · 2025-04-01
>
> Dear reviewer, we are delighted that you like our score-based reinforcement learning post-training of the one-step diffusion model. We appreciate your valuable suggestions. In the following paragraphs, we will address your concerns one by one.
>
> **Q1**. Clarify the differences between Diff-Instruct-star, DMD, and other approaches.
>
> **A1**. We are sorry for the confusion. In one-step diffusion distillation/post-training literature, there are typically two types of training criteria.
>
> + The first type is trajectory-based methods, such as Progressive distillation, consistency distillation, and other variants.
>
> + The second type is **distribution matching based methods**, which aim to minimize some probability divergence between one-step model distribution and teacher diffusion model distribution. Among them, **Diff-Instruct** first proposes one-step diffusion distillation by minimizing the integral KL divergence. **DMD** and **DMD2** generalize the KL divergence minimization by incorporating a regression loss and a GAN loss. **SiD**[4] studies the minimize the one-step distillation by minimizing Fisher divergence instead of KL. Later, **SIM** unifies the one-step diffusion distillation by minimizing general score-based divergences (including Fisher divergence) with detailed theories that prove the gradient equivalence between SIM loss and the underlying divergence.
>
> + **Post-training**: The **reward-guided LCM**  first combines a reward model with consistency loss to train few-step diffusion models. **DI++** then studied the post-training of the one-step model by introducing a KL divergence-PPO framework and achieved a very strong human preference quality. Our DIstar is inspired by DI++, by differs in (1) theoretical and empirical study the post-training via score-based-PPO instead of KL-based PPO; (2) introducing novel techniques that successfully scales the post-training of one-step diffusion model at 1024x1024 resolution, and outperforms current SoTA diffusion model (FLUX-dev) with only 1-step (1.8\%) inference costs. This surprisingly strong performance also gives us the inspiration for our title **Small one-step model beats large diffusion with score-based post-training**.
>
> In conclusion, DIstar follows the Occam's Razor principle: achieving better human-preference quality than open-sourced SoTA FLUX-dev with minimal losses (reward and general score-divergence).
>
> **Q2**. Discussions on why the score-based divergence is potentially better in diversity-preserving than KL.
>
> **A2**. We appreciate your good question! We will answer your question from both theoretical and empirical perspectives.
>
> **Theoretical perspective**: If we use $p_ r$ to represent reference distribution and $p_{g}$ the one-step generator distribution. The KL divergence between $p_ g$ and $p_ r$ is defined as:
> $D_ {KL}(p_ g||p_ r) = \mathbb{E}_ {x\sim p_g}\log \frac{p_ g (x)}{p_ r (x)}$
> while the general score-based divergence with a form $\mathcal{D}_ {SD}(p_ g,p_ d) = \mathbb{E}_ {x\sim \pi} d(\nabla_ x \log p_ g (x) - \nabla_ x \log p_ d (x))$.
> KL divergence is long believed to be bothered by mode-collapse issues because the likelihood ratio $\frac{p_ g (x)}{p_ r (x)}$ will be ill-defined if the  $p_ g$ and $p_ d$ have misaligned density support. However, the score-based divergence does not have such a "ratio" that will be ill-posed and, therefore, is safe in the case when $p_ g$ and $p_ r$ have misaligned support. Besides, the distance function $d()$ of general score-based divergence also brings flexibility in defining divergences to properly measure the difference between two distributions in a robust magnitude. For instance, our used Pseudo-Huber distance (line 259)  has a self-normalization form, which also helps stabilize the gradient of score-divergence to a unit scale, potentially further helping address the mode-collapse issue.
>
> **Empirical perspective**: We conduct a new experiment on one-step diffusion distillation on the CIFAR10 unconditional. The FID and Recall of KL and score-divergence model are (3.70,0.52) and (2.01,0.60). This shows that distilling using KL divergence leads to both worse FID and Recall.
>
> **Q3**. It would be good if the authors could present more qualitative examples comparing to KL-based models, such as DI++-SDXL, to verify the motivation of the method.
>
> **A3**. In **Fig. 4** in the appendix of the paper, we qualitatively compare the DIstar model with DI++, SIM, Diff-Instruct, DMD2, SDXL, and SDXL-DPO models. We clearly find that images generated by the DIstar model show better aesthetic quality, with gentle lights and colors. We also put more qualitative comparisons between DIstar-SDXL-DPO and DMD-SDXL-DPO in this anonymous link: https://anonymous.4open.science/r/distar_anonymous-E007/distar_dmd.png.
>
> **We appreciate your valuable review. We sincerely hope our response has resolved your concerns. If you still have any concerns, please let us know.**

---

### Decision · Program_Chairs · 2025-05-01

**Decision:**

Accept (poster)

**Comment:**

This paper has received all acceptances in the final recommendations. The authors introduce Diff-Instruct*, a novel post-training method designed to align one-step text-to-image generative models with human preferences. While the initial reviews highlighted concerns regarding unclear explanations and the absence of qualitative analysis, the authors' rebuttal effectively addressed several of these points. This response led to a consensus among the reviewers, ultimately resulting in the Area Chair's decision to accept the paper.